# Exact and Rich Feature Learning Dynamics of Two-Layer Linear Networks

Wei Huang[1], Wuyang Chen[2], Zhiqiang Xu[3], Zhangyang Wang[4], Taiji Suzuki[5,1]
[1]RIKEN AIP, [2]Simon Fraser University,
[3]MBZUAI, [4]University of Texas at Austin, [5]University of Tokyo
`wei.huang.vr@riken.jp, wuyang@sfu.ca`
zhiqiang.xu@mbzuai.ac.ae, atlaswang@utexas.edu, taiji@mist.i.u-tokyo.ac.jp

Deep neural networks exhibit rich training dynamics under gradient descent updates. The root of this phenomenon is the non-convex optimization of deep neural networks, which is extensively studied in recent theory works. However, previous works did not consider or only considered a few gradient descent steps under non-asymptotic manner, resulting in an incomplete characterization of the network's stage-wise learning behavior and the evolutionary trajectory of its parameters and outputs. In this work, we characterize how a network's feature learning happens during training in a regression setting. We analyze the dynamics of two quantities of a two-layer linear network: the projection of the first layer's weights onto the feature vector, and the weights in the second layer. The former indicates how well the network fits the feature vector from the input data, and the latter stands for the magnitude learned by the network. More importantly, by formulating the dynamics of these two quantities into a non-linear system, we give the precise characterization of the training trajectory, demonstrating the rich feature learning dynamics in the linear neural network. Moreover, we establish a connection between the feature learning dynamics and the neural tangent kernel, illustrating the presence of feature learning beyond lazy training. Experimental simulations corroborate our theoretical findings, confirming the validity of our proposed conclusion.

## 1. Introduction

Deep learning achieves huge accomplishments in solving diverse machine learning problems [1–3]. Despite the success in practical applications, the underlying mechanism of deep learning is still not well understood, leaving a gap between fundamental understanding and real-life practice of deep neural networks. The over-parameterized model, which consists of more parameters than the number of training samples, can be trained to memorize all the training data (converging to zero training loss) even with the existence of random noises. Meanwhile, over-parameterized models can still exhibit generalization on unseen test data, contradicting classical analysis on the bias-variance trade-off in statistical learning theory. This phenomenon has been observed in numerous previous works [4, 5], and intensively studied by the theory community [6–15].

One promising line of research to analyze deep networks is to exploit the connection between over-parameterized neural networks and kernel machines. People find the similarity between kernel machines and deep networks: both can memorize and interpolate the training data [16, 17]. Compared with deep neural networks, kernel machines have the advantage of more solid theory in their training and generalization [18, 19] and better interpretability [20]. Recent studies show that infinitely wide neural networks trained with gradient descent can be approximated by its linearized version in parameter space [21, 22] and thus converging to a kernel regression solution with a kernel called the neural tangent kernel (NTK) in function space [23–27]. However, the exact equivalence between neural networks and NTKs breaks for finite width networks [28–30] or initialization with large variance [31, 32]. Furthermore, in the network training regime with an approximately static kernel, also referred to as the "lazy training" regime [31], network parameters remain close to their

initialization during training and cannot explain the adaptation of deep networks and its learned features to the training data [14, 33], which is believed to be the key for deep networks to outperform kernel methods [34–36].

In this work, we establish a novel connection between feature learning and neural network optimization through the lens of weight decomposition in the direction of the feature vector, considering a non-asymptotic perspective. For a linear two-layer network, we analyze the dynamics of two quantities: 1) the projection of the first layer weight to the target feature vector $\mathbf{X}^\top \mathbf{y}$ (dubbed "feature alignment"), where $\mathbf{X}$ and $\mathbf{y}$ are input data and ground truth; 2) the weight magnitude in the second layer (dubbed "network magnitude"). Disentangled analysis of these two quantities helps us precisely characterize the network's behavior during gradient descent and the reason behind it. The "feature alignment" indicates how well the network fits the noisy training data, and the "network magnitude" identifies the magnitude of the network's output. By formulating these two critical quantities into a non-linear dynamic system, we successfully track the trajectory of weights during the gradient descent training in a regression setting, thereby capturing the rich and precise feature learning dynamics inherent in linear neural networks.

Our results indicate a two-stage dynamics of feature learning. In the early training phase, the network does not actively minimize the loss with its network magnitude staying at a low level. We find the network actively learns the directions of the target feature vector and increases its "feature alignment". In the second phase, the output magnitude increases, and the network follows the fixed direction of the target feature vector and keeps learning gradient magnitudes. Finally, we establish a connection between feature learning dynamics and neural tangent kernel (NTK) [23]. In the initial stage, the neural network predominantly focuses on mastering the direction of the feature vector, causing the NTK to maintain its norm while evolving towards a kernel that is aligned with the new target. In the latter stage, while the direction is retained, the network amplifies its magnitude, leading to an increase in the NTK's norm while maintaining the structure of the target-aligned kernel. Collectively, the transformations in the NTK underscore that the neural network is engaged in feature learning, standing in contrast to the dynamics observed in lazy training [31].

Our contributions are summarized below:

- We precisely characterize two training phases in a two-layer network from a non-asymptotic perspective. The first layer weight learns to align with the target and the loss remains high in the first phase. After that, the weight amplifies its projection magnitude in the learned direction and the loss is actively minimized in the second phase.

- We determine the training time step that separates the two training phases by solving the non-linear system of feature alignment and network magnitude. Based on our analysis of the stage-wise dynamics, we showcase rich and precise dynamics of feature learning beyond lazy training in a regression setting with finite width. A comparison between our work and existing works can be found in Table 1.

- We conduct experiments on both synthetic and real-world datasets. The simulation results further verify our theoretical findings.

## 2. Related Work

**Feature learning**  To explain the gap between kernel machines and networks in practice, people recently started analyzing beyond the lazy-training regime and trying to characterize the network's feature learning behavior. Related works in this line can be summarized into three categories: (i) The first direction is to calculate finite width corrections to the infinite width limit and capture non-Gaussian processes in practical networks [29, 37–41]. (ii) Secondly, people try to allow feature learning during gradient descent by altering network parameterizations and learning rates, even at infinite width. The mean field limit [42–44] emerges as alternative parameterizations where feature learning is significant. (iii) A third line of works unifies the network parametrizations and proposes the Maximal Update Parametrization ($\mu P$), which admits maximal feature learning and enables ef-

ficient hyper-parameter search and transfer [45]. People also study the phase diagram of networks regarding their initialization variance and learning rates and identify linear/critical/condensed regimes of different feature learning behaviors [46, 47]. Finally, empirical studies also gain insight into the gap between lazy training and feature learning regimes of deep networks. These works observe the evolution of NTK spectrum, kernel similarity, and the network's output approximated by kernel regression [32, 48–54].

**Phases in training deep networks** Behaviors of the deep network during gradient descent training show phase transitions. The information bottleneck (IB) theory [55, 56] claims the transition from the initial fitting phase to a subsequent compression phase during gradient descent training. Recent works [8, 57] demonstrated that in the early phase of training, the curve of a finite-width neural network can be approximated by a linear model. Moreover, [58] observed that at the end of the early training phase, gradients span a low-dimensional subspace. [59–61] showed that an initial large learning rate can benefit late-time generalization performance. Empirical observations on the evolution of NTK kernel similarities and generalization were also studied by [48, 62]. More rigorously, [63] tried to explain the different learning phases of NTK alignment and training loss. In the low-rank matrix recovery, [64] also achieved multiple-phase training dynamics: an early alignment phase, followed by a learning on NTK spectral magnitude, and finally refinement phase. Their results share qualitative similarities with our analysis of linear networks. More recently, [65] conducted a complete theoretical characterization of the training process of a two-layer ReLU network trained by Gradient Flow on linearly separable data. They revealed four different phases from the whole training process showing a general simplifying-to-complicating learning trend. In our work, we give the precise characterization of stage-wise dynamics in a regression setting.

# 3. Preliminaries

## 3.1. Notation

We use lower and capital bold-faced letters for vectors and matrices, respectively, otherwise representing scalar. We use $\|\cdot\|_2$ to denote the Euclidean norm of a vector or the spectral norm of a matrix while denoting $\|\cdot\|_F$ as the Frobenius norm of a matrix. Moreover, for any positive semi-definite (PSD) matrix $\mathbf{A} \in \mathbb{R}^{d \times d}$ and any vector $\mathbf{v} \in \mathbb{R}^d$, we denote $\|\mathbf{v}\|_{\mathbf{A}} = \sqrt{\mathbf{v}^\top \mathbf{A} \mathbf{v}}$. Let $\mathbf{I}_d$ be the identity matrix with the dimension of $\mathbb{R}^{d \times d}$. We denote $[n] = \{1, 2, \ldots, n\}$. For a random variable $Z$, we denote by $\|Z\|_{\psi_2}$ and $\|Z\|_{\psi_1}$ the sub-Gaussian and sub-exponential norms of $Z$, respectively. Given two sequences $\{a_n\}$ and $\{b_n\}$, we use standard asymptotic notations $O(\cdot)$, $o(\cdot)$, $\Omega(\cdot)$, $\omega(\cdot)$, and $\Theta(\cdot)$ to describe the limiting behavior between them. In particular, we denote by $a_n = O(b_n)$ that there exists a positive real number $C_1$ and a positive integer $N$ such that $|a_n| \le C_1 |b_n|$ for all $n \ge N$. Similarly, we write $a_n = \Omega(b_n)$ if there exists $C_2 > 0$ and $N > 0$ such that $|a_n| > C_2 |b_n|$ for all $n \ge N$. As a result, we say $a_n = \Theta(b_n)$ if $a_n = O(b_n)$ and $a_n = \Omega(b_n)$. Besides, if $\lim_{n \to \infty} |a_n/b_n| = 0$, we say $a_n = o(b_n)$; we write $a_n = \omega(b_n)$ if $\lim_{n \to \infty} |a_n/b_n| = \infty$.

## 3.2. Data Model

We consider a random feature generation model. Define the covariance for data feature $\boldsymbol{\Sigma} = \mathbb{E}_{\mathbf{x}}[\mathbf{x}\mathbf{x}^\top] \in \mathbb{R}^{d \times d}$, where $\boldsymbol{\Sigma}$ is a positive definite matrix with eigenvalue decomposition $\boldsymbol{\Sigma} = \mathbf{U}\boldsymbol{\Lambda}\mathbf{U}^\top$. In particular, $\boldsymbol{\Lambda} = \text{diag}\{\lambda_1, \lambda_2, \cdots, \lambda_d\} \in \mathbb{R}^{d \times d}$ and $\mathbf{U} \in \mathbb{R}^{d \times d}$ is an orthogonal matrix consisting of the eigenvectors of $\boldsymbol{\Sigma} \in \mathbb{R}^{d \times d}$. Then the data is generated by $\mathbf{x}_i = \mathbf{U}\boldsymbol{\Lambda}^{\frac{1}{2}}\mathbf{z}_i$, where $\mathbf{z}_i \in \mathbb{R}^d$ has components that are independent $\sigma_x^2$-subgaussian with zero mean. This ensures that $\mathbf{x}_i \in \mathbb{R}^d$ has mean zero and covariance matrix $\boldsymbol{\Sigma}$. We further assume a linear teacher (ground truth) model

$$y_i = f^*(\mathbf{x}_i) + \epsilon_i = \langle \mathbf{x}_i, \boldsymbol{\beta} \rangle + \epsilon_i, \tag{1}$$

where $\boldsymbol{\beta} \in \mathbb{R}^d$ is some unknown but given feature vector, and $\epsilon_i \in \mathbb{R}$ is noise term that is i.i.d. sub-Gaussian with mean zero and variance $\sigma_y^2$. Then the training set $(\mathbf{X}, \mathbf{y}) \in (\mathbb{R}^{n \times d}, \mathbb{R}^n)$ of $n$ i.i.d. sample are generated from Equation (3.2) and Equation (1).

### 3.3. Neural Network Model and Gradient Descent

In this work, we consider a two-layer linear neural network $f$ which can be expressed as $f(\mathbf{X}, \mathbf{W}, \mathbf{v}) = \frac{1}{m} \sum_{r=1}^{m} v_r \cdot \mathbf{X} \mathbf{w}_r$, where $\mathbf{X} \in \mathbb{R}^{n \times d}$ is the input data, $\mathbf{W} \equiv [\mathbf{w}_1, \mathbf{w}_2, \cdots \mathbf{w}_m]^\top \in \mathbb{R}^{d \times m}$ is the weight matrix at the first (input) layer, and $\mathbf{v} \equiv [v_1, v_2, \cdots, v_m]^\top \in \mathbb{R}^m$ is the weight vector at the second (output) layer. The width of the neural network is denoted by $m$.

We initialize the parameters randomly in this work. In particular, we have, for $r \in [m]$, $\mathbf{w}_r \sim \mathcal{N}(\mathbf{0}, \sigma_0^2 \cdot \mathbf{I}_d)$, $v_r \sim v_0 \cdot \text{uniform}(\{-1, 1\})$, where $\sigma_0$ controls the magnitude of initialization for the weight in the first layer and $v_0$ controls the magnitude of second layer weight initialization. We then optimize the weight in both layers through gradient descent on the empirical loss function $L(f(\mathbf{X}, t), \mathbf{y}) = \frac{1}{2n} \|f(\mathbf{X}, t) - \mathbf{y}\|_2^2$. Given the learning rate $\eta$, for $r \in [m]$, the gradient descent update rule of the neural network can be expressed as:

$$
\begin{aligned}
\mathbf{w}_r(t+1) &= \mathbf{w}_r(t) + \frac{\eta}{mn} \cdot v_r(t) \cdot \mathbf{X}^\top \mathbf{y} - \frac{\eta}{m^2 n} \cdot v_r(t) \cdot \mathbf{X}^\top \mathbf{X} \mathbf{W}(t) \mathbf{v}(t), \\
v_r(t+1) &= v_r(t) + \frac{\eta}{mn} \cdot \mathbf{w}_r^\top(t) \mathbf{X}^\top \mathbf{y} - \frac{\eta}{m^2 n} \cdot \mathbf{w}_r^\top(t) \mathbf{X}^\top \mathbf{X} \mathbf{W}(t) \mathbf{v}(t).
\end{aligned}
\tag{2}
$$

### 3.4. Neural Tangent Kernel

Given a two-layer linear neural network $f$, it is found that the output function with gradient flow admits the following dynamics,

$$
\frac{df(\mathbf{X}; t)}{dt} = \frac{\partial L(f(\mathbf{X}; t), \mathbf{y})}{\partial f(\mathbf{X}; t)} \mathbf{\Theta}(\mathbf{X}, \mathbf{X}; t),
\tag{3}
$$

where $L(f(\mathbf{X}; t), \mathbf{y})$ is the empirical loss function, and in this work we adopt the squared loss. Moreover, $\mathbf{\Theta}(\mathbf{X}, \mathbf{X}; t)$ is the neural tangent kernel (NTK) that is defined as follows:

**Definition 3.1** (NTK). *The tangent kernels associated with the output function $f(\mathbf{X}; t)$ at parameters $\boldsymbol{\theta} \triangleq (\mathbf{W}, \mathbf{v})$ can be expressed as*

$$
\mathbf{\Theta}(\mathbf{X}, \mathbf{X}; t) = \frac{\partial f(\mathbf{X}, \boldsymbol{\theta}; t)}{\partial \boldsymbol{\theta}} \left( \frac{\partial f(\mathbf{X}, \boldsymbol{\theta}; t)}{\partial \boldsymbol{\theta}} \right)^\top \in \mathbb{R}^{n \times n}.
\tag{4}
$$

The neural tangent kernel theory [23] states that the NTK will converge to a deterministic kernel at initialization and during gradient descent training, thus providing a convergence guarantee for the neural network in the infinite-width limit. In contrast to the neural tangent kernel theory where the width is set to be disproportionally large, this work studies feature learning and will show that the neural network will learn a new target-aligned kernel.

## 4. Main Results

In this section, we demonstrate our main results regarding the optimization of a two-layer neural network under gradient descent.

Our results are based on the following conditions on dimension $d$, sample size $n$, neural network width $m$, learning rate $\eta$, initialization scale $\sigma_0$ $v_0$, trace two-norm and F-norm of input covariance matrix $\text{tr}(\mathbf{\Sigma}), \|\mathbf{\Sigma}\|_2, \|\mathbf{\Sigma}\|_F$, and noise variance $\sigma_y^2$.

**Assumption 4.1.** *Let $\epsilon > 0$. We assume that*

*(1) The width of the hidden layer follows $m = \Omega(\log(2n))$.*

*(2) The first layer weight initialization satisfies $\sigma_0 \leq \min \left\{ \frac{C_1}{\sqrt{nd} \|\overline{\boldsymbol{\beta}}\|_2^2 \text{tr}(\mathbf{\Sigma})}, \frac{\sqrt{\epsilon}}{n^{1/4} \|\overline{\boldsymbol{\beta}}\|_2^{1/2} \text{tr}(\mathbf{\Sigma})^{1/4}} \right\}$.*

*(3) The second layer weight initialization satisfies $v_0 \leq \frac{C_2}{\sqrt{n \|\overline{\boldsymbol{\beta}}\|_2^2 \text{tr}(\mathbf{\Sigma})}}$.*

(4) *The learning rate $\eta$ satisfies $\eta = O(\frac{\sqrt{nm}}{\sqrt{\mathrm{tr}(\boldsymbol{\Sigma})}\|\overline{\boldsymbol{\beta}}\|_2})$.*

(5) *The noise variance follows $\sigma_y^2 \leq 8n\|\overline{\boldsymbol{\beta}}\|_2^2$.*

(6) $\mathrm{tr}(\boldsymbol{\Sigma}) \geq \max\{C_2, \sqrt{\epsilon}\|\overline{\boldsymbol{\beta}}\|_2\}\sigma_x^2(n \cdot \|\boldsymbol{\Sigma}\|_2 + \sqrt{n} \cdot \|\boldsymbol{\Sigma}\|_F)$ *and* $\mathrm{tr}(\boldsymbol{\Sigma})\|\overline{\boldsymbol{\beta}}\|_2^2 = \Omega(1)$.

*Here we define $\|\overline{\boldsymbol{\beta}}\|_2 \triangleq \|\boldsymbol{\beta}\|_{\boldsymbol{\Sigma}} = \sqrt{\boldsymbol{\beta}^\top \boldsymbol{\Sigma}\boldsymbol{\beta}}$. Besides, $C_1$, $C_2$, and $C_3$ are absolute constants.*

A few remarks on Condition 4.1 are in order. (1) We require the neural network width to be at least polylogarithmic in the sample size $n$ to ensure some statistical properties of the weight initialization to hold with probability at least $1-O(1/n^2)$. This is a mild condition compared to the neural tangent kernel theory [66, 67]. (2,3) The conditions on initialization strength $\sigma_0$ and $v_0$ are to ensure that there are two stage dynamics in the whole training process. (4) The learning rate is chosen such that the gradient descent can effectively minimize the training loss. (5) The condition on noise variance is to ensure that the noise is small enough compared to the feature part. (6) The final condition on $\mathrm{tr}(\boldsymbol{\Sigma})$ ensures that $\mathbf{X}\mathbf{X}^\top$ is close to a scaled identity matrix.

## 4.1. A Two-Stage Dynamics for Feature Learning

The core of our analyses and results lies in a two-stage behavior of the training dynamics in the two-layer neural network trained by gradient descent. Intuitively, the initial neural network weights are small enough so that the output neural network at initialization has a small and negligible magnitude compared to its ground truth: $f(\mathbf{x}_i) \ll y_i$ for all $i \in [n]$. This is guaranteed under Assumption 4.1. Then by the gradient descent update rule (2) with a more compact format, we have

$$
\begin{aligned}
\mathbf{w}_r(t+1) &= \mathbf{w}_r(t) + \frac{\eta}{n} \cdot \left(\frac{\partial f(\mathbf{X})}{\partial \mathbf{w}_r(t)}\right)^\top \mathbf{y} - \frac{\eta}{n} \cdot \left(\frac{\partial f(\mathbf{X})}{\partial \mathbf{w}_r(t)}\right)^\top f(\mathbf{X}, t), \\
v_r(t+1) &= v_r(t) + \frac{\eta}{n} \cdot \left(\frac{\partial f(\mathbf{X})}{\partial v_r(t)}\right)^\top \mathbf{y} - \frac{\eta}{n} \cdot \left(\frac{\partial f(\mathbf{X})}{\partial v_r(t)}\right)^\top f(\mathbf{X}, t).
\end{aligned}
\tag{5}
$$

It is observed that in the *initial stage*, the third term on the right-hand side of Equation (5) is negligible compared to the second term. This observation leads to a simplified, yet insightful, gradient descent rule for the weights in the neural network $\mathbf{w}_r(t+1) \approx \mathbf{w}_r(t) + \frac{\eta}{nm} \cdot v_r(t) \cdot \mathbf{X}^\top \mathbf{y}$. This equation suggests that the gradient descent iterate $\mathbf{w}_r(t)$ is essentially a linear combination of its random initialization $\mathbf{w}_r(0)$ and the feature vector $\mathbf{X}^\top \mathbf{y}$ derived from the training data in the initial stage.

Motivated by this observation, we define a coefficient to measure the projection of the first-layer weights onto $\mathbf{X}^\top \mathbf{y}$ during the gradient descent training:

$$
\rho_r(t) \triangleq \langle \mathbf{w}_r(t) - \mathbf{w}_r(0), \mathbf{X}^\top \mathbf{y}\rangle,
\tag{6}
$$

where $\rho_r(0) = 0$. In particular, $\rho_r(t)$ characterizes the progression of learning the feature vector $\mathbf{X}^\top \mathbf{y}$. Evidently, based on the definition (6), for some iteration $t$, we observe the following:

1 If all $\rho_r(t)$ values are sufficiently large and $v_r(t)$ values are relatively small such that $f(\mathbf{X}, t)$ is small compared to $\mathbf{y}$, then the neural network effectively learns the features.

2 Conversely, if all $\rho_r(t)$ values are small but $v_r(t)$ values are large such that $f(\mathbf{X}, t)$ is comparable to $\mathbf{y}$, the neural network is performing lazy training, reducing to the neural tangent kernel regime as discussed in references such as [31] and [23].

Equation (6) offers us an approach to study the dynamics of feature learning. By incorporating it into the gradient descent update rule (2), we obtain the following updates:

$$
\rho_r(t+1) = \rho_r(t) + \frac{\eta v_r(t)}{mn}\langle \mathbf{X}^\top \mathbf{y}, \mathbf{X}^\top \mathbf{y}\rangle - \frac{\eta v_r(t)}{m^2 n}\langle \mathbf{X}^\top \mathbf{X}\mathbf{W}(t)\mathbf{v}(t), \mathbf{X}^\top \mathbf{y}\rangle,
\tag{7}
$$

$$
v_r(t+1) = v_r(t) + \frac{\eta \rho_r(t)}{mn} + \frac{\eta\langle \mathbf{w}_r(0), \mathbf{X}^\top \mathbf{y}\rangle}{mn} - \frac{\eta}{m^2 n}\langle \mathbf{w}_r(t), \mathbf{X}^\top \mathbf{X}\mathbf{W}(t)\mathbf{v}(t)\rangle.
\tag{8}
$$

Table 1: Comparison with existing works. We highlight the unique property of our results in `color` .

| Research works | Setting | Model | Data | Training dynamics |
|---|---|---|---|---|
| [13] | Classification | 2-layer non-linear | $\mathbf{x} = \mathbf{z} + y\boldsymbol{\beta}$ | Feature learning |
| [14] | Classification | 2-layer CNN | $\mathbf{x} = [y\boldsymbol{\beta}, \mathbf{z}]$ | 2-stage feature learning |
| [65] | Classification | 2-layer ReLU | Linear separable | 4-stage dynamics |
| [68] | Classification | 2-layer ReLU | XOR | Population loss dynamics |
| [69] | Regression | Deep linear | $\lambda_{\min}(\mathbf{X}\mathbf{X}^\top) > 0$ | Lazy training |
| [70] | Regression | 2-layer linear | Full rank | Balance-induced |
| [71] | Regression | 2-layer ReLU | Sparse coding | Infinitesimal initialization & Initial stage |
| [72] | Regression | 2-layer ReLU | Gaussian isotropic | Special gradient descent |
| [11] | Regression | 2-layer linear | $\mathbf{x} = \mathbf{U}\boldsymbol{\Lambda}^{\frac{1}{2}}\mathbf{z}$ | N/A |
| [73] | Regression | 2-layer ReLU | Correlated inputs | Balance-induced |
| [74] | Regression | 2-layer ReLU | Orthogonal family | Gradient flow & balanced initialization |
| [75] | Regression | 2-layer non-linear | Gaussian isotropic | Population loss dynamics |
| [15] | Regression | 2-layer non-linear | Gaussian | One gradient step |
| This work | Regression | 2-layer linear | $\mathbf{x} = \mathbf{U}\boldsymbol{\Lambda}^{\frac{1}{2}}\mathbf{z}$ | Exact non-asymptotic training dynamics |

Equations (7) and (8) elucidate how $\rho_r(t)$ and $v_r(t)$ for all $r \in [m]$ evolve during the gradient descent training process; hence, we refer to them as the feature learning dynamics equations.

Based on Assumption 4.1 we present our principal findings on the feature learning dynamics, as represented by Equations (7) and (8), throughout the entirety of the training process in the subsequent theorem.

**Theorem 4.1.** *Suppose $\epsilon > 0$. Under Assumption 4.1, with probability at least $1 - O(1/n^2)$ over the randomness of training data and weight initialization, for any $\epsilon > 0$, there exists $t = \Omega\left(\frac{m\sqrt{n}\log(1/v_0)}{\eta\sqrt{\mathrm{tr}(\boldsymbol{\Sigma})}\|\bar{\boldsymbol{\beta}}\|_2} + \frac{m\sqrt{n}}{\eta\|\bar{\boldsymbol{\beta}}\|_2\sqrt{\mathrm{tr}(\boldsymbol{\Sigma})}}\log(\frac{n\|\bar{\boldsymbol{\beta}}\|_2\mathrm{tr}(\boldsymbol{\Sigma})}{\epsilon})\right)$ such that $L(t) \le 4\epsilon$. Furthermore, the entire dynamics can be described as a two-stage process:*

- *In the initial stage, $0 \le t \le t_1 \triangleq \frac{C_1 m\sqrt{n}\log(1/v_0)}{\eta\sqrt{\mathrm{tr}(\boldsymbol{\Sigma})}\|\bar{\boldsymbol{\beta}}\|_2}$. By the end of this stage, $v_r(t_1) = \Theta(1/\sqrt{\mathrm{tr}(\boldsymbol{\Sigma})n\|\bar{\boldsymbol{\beta}}\|_2^2})$ and $\rho_r(t_1) = \Theta(1)$, for all $r \in [m]$. Here we denote $\gamma = \mathbf{y}^\top\mathbf{X}\mathbf{X}^\top\mathbf{y}$.*

- *From $t_1$, the loss function converges to $L(t) \le \epsilon$ until $t = \Omega\left(\frac{m\sqrt{n}\log(1/v_0)}{\eta\sqrt{\mathrm{tr}(\boldsymbol{\Sigma})}\|\bar{\boldsymbol{\beta}}\|_2} + \frac{m\sqrt{n}}{\eta\|\bar{\boldsymbol{\beta}}\|_2\sqrt{\mathrm{tr}(\boldsymbol{\Sigma})}}\log(\frac{n\|\bar{\boldsymbol{\beta}}\|_2\mathrm{tr}(\boldsymbol{\Sigma})}{\epsilon})\right)$.*

Theorem 4.1 demonstrates the convergence of the linear network with gradient descent training in a non-asymptotic manner. In particular, the training dynamics consist of a two-stage behavior for feature learning dynamics. In the first stage, the neural network leverages the period when the output function remains relatively small to capture the data feature vectors effectively. At the same time, the neural tangent kernel aligns closely with a new target-aligned kernel, facilitating the transition to the next stage of training. The second stage fully incorporates the exact gradient descent rules (2). Due to the insights gained in Stage 1, a significant scale difference is maintained between $\rho_r(t)$ and $v_r(t)$, which is crucial for the analysis. This stage ensures that the neural network kernel aligns in the direction of the ground truth while its magnitude increases progressively.

In summary, Theorem 4.1 characterizes feature learning dynamics until convergence. It shows that, under Assumption 4.1, the neural network is able to learn feature from the input and achieve a small training loss under gradient descent training. To demonstrate the contribution of this work, we make a comparison with existing related works in Table 1. We emphasize the characterization of full feature learning dynamics in a regression setting and a non-asymptotic manner.

# 5. Proof Sketch

In this section, we outline the primary challenges encountered in the study of feature learning dynamics and discuss the essential techniques employed in our proofs to address these challenges. Comprehensive proofs of all results are provided in the appendix.

## 5.1. Iterative Analysis of the Gradient Descent

In order to study the learning process based on non-convex optimization, we propose a key technique by introducing $\rho_r(t)$, as illustrated in Equation (6). This approach allows for an iterative examination of the weight vector's projection in the direction of the feature vector $\mathbf{X}^\top \mathbf{y}$. As a result, we transform the dynamics of the weights into the tracking dynamics of $\rho_r(t)$ and $v_r(t)$ governed by Equations (7) and (8). In particular, we utilize a two-phase analysis approach to disentangle the intricate dynamics involved in the learning process.

**Phases 1: Feature learning** In the initial stage of training, when the weights are initialized to be relatively small (i.e., when $\sigma_0$ and $v_0$ are sufficiently small), the impacts of the output function $f(\mathbf{x}_i)$ and $\langle \mathbf{w}_r(0), \mathbf{X}^\top \mathbf{y} \rangle$ on the training dynamics are approximately negligible. Consequently, the feature learning dynamics, as described by Equations (7) and (8), simplify to:

$$\rho_r(t+1) \approx \rho_r(t) + \frac{\eta}{nm} \cdot v_r(t) \cdot \gamma, \quad v_r(t+1) \approx v_r(t) + \rho_r(t) + \frac{\eta \langle \mathbf{w}_r(0), \mathbf{X}^\top \mathbf{y} \rangle}{mn}. \tag{9}$$

Here we denote $\gamma \triangleq \mathbf{y}^\top \mathbf{X} \mathbf{X}^\top \mathbf{y}$. Equation (9) is a linear dynamics system which enables us to obtain an explicit solution of dynamics. It is evident that the signs of $\rho_r(t)$ and $v_r(t)$ for $t > 1$ are determined by the initial values of $\langle \mathbf{w}_r(0), \mathbf{X}^\top \mathbf{y} \rangle$ and $v_r(0)$. Consequently, for each $r \in [m]$, $\rho_r(t)$ and $v_r(t)$ maintain consistent signs throughout the training. Furthermore, the projection of the first-layer weight $\rho_r(t)$ behaves like a $\sqrt{\gamma} \sinh(\frac{\eta \sqrt{\gamma}}{nm} t)$, while the second-layer weight $v_r(t)$ aligns with the function $\cosh(\frac{\eta \sqrt{\gamma}}{nm} t)$. Over time, the difference between $\rho_r(t)$ and $\sqrt{\gamma} v_r(t)$ diminishes.

To **accurately** capture the dynamics in the first stage, we track the dynamics of $\rho_r(t)$ and $v_r(t)$ governed by Equations (7) and (8). Two techniques are employed to control the randomness in the initialization of the first-layer weight $\mathbf{w}_r(0)$, specifically from the term $\langle \mathbf{w}_r(0), \mathbf{X}^\top \mathbf{y} \rangle$ and the gradient descent term involving $\mathbf{X}^\top \mathbf{X} \mathbf{W}(t) \mathbf{v}(t)$. First, by ensuring a sufficiently small $\sigma_0$, the perturbation due to weight initialization can be mitigated. Second, by setting $\mathrm{tr}(\mathbf{\Sigma})$ larger than $\sigma_x^2(n\|\mathbf{\Sigma}\|_2 + \sqrt{n}\|\mathbf{\Sigma}\|_F)$, we can control the variance of the eigenvalue distribution of $\mathbf{X} \mathbf{X}^\top$ according to the following Lemma:

**Lemma 5.1** (Lemma A.4 in [76])**.** *With probability at least* $1 - n^{-2}$, *we have,* $\|\mathbf{X}\mathbf{X}^\top - \mathrm{tr}(\mathbf{\Sigma}) \cdot \mathbf{I}\|_2 \leq C\sigma_x^2(n\|\mathbf{\Sigma}\|_2 + \sqrt{n}\|\mathbf{\Sigma}\|_F)$, *where* $C$ *is an absolute constant.*

Therefore, $\mathbf{X}^\top \mathbf{X} \mathbf{W}(t) \mathbf{v}(t)$ tends to align closely with the direction of $\mathbf{X}^\top \mathbf{y}$ when $\mathbf{W}(t)\mathbf{v}(t)$ sufficiently evolves in the direction of $\mathbf{X}^\top \mathbf{y}$. To maintain the influence of $\mathbf{X}^\top \mathbf{X} \mathbf{W}(t)\mathbf{v}(t)$ within manageable bounds, we establish a threshold for the first stage, ensuring that $\rho_r(t)$ reaches $\Theta(1)$. The following lemma summarizes our main conclusion at Stage 1 for feature learning:

**Lemma 5.2.** *Under Assumption 4.1, there exists a time step* $t_1 = \frac{C_1 m \sqrt{n} \log(1/v_0)}{\eta \sqrt{\mathrm{tr}(\mathbf{\Sigma})}\|\overline{\boldsymbol{\beta}}\|_2}$, *with probability at least* $1 - O(n^{-2})$, *such that: (i) for all* $r \in [m]$, *we have* $\rho_r(t_1) = \Theta(1)$. *(ii) for all* $r \in [m]$, *we have* $v_r(t_1) = \Theta(1/\sqrt{n\|\overline{\boldsymbol{\beta}}\|_2^2 \mathrm{tr}(\mathbf{\Sigma})})$. *(iii)* $\left\| \Theta(\mathbf{X}, \mathbf{X}, t_1) - \frac{1}{m\gamma} \mathbf{X} \left( \mathbf{I} + \frac{1}{\gamma} \mathbf{X}^\top \mathbf{y} \mathbf{y}^\top \mathbf{X} \right) \mathbf{X}^\top \right\|_2 \leq \frac{C_2}{mn^3 \|\overline{\boldsymbol{\beta}}\|_2^6 \mathrm{tr}(\mathbf{\Sigma})}$. *Here* $C_1$ *and* $C_2$ *are absolute constants.*

Lemma 5.2 establishes that by the conclusion of Stage 1, all instances of $\rho_r(t)$ reach $\Theta(1)$ and all instances of $v_r(t)$ arrive at $\Theta(1/\sqrt{n\|\overline{\boldsymbol{\beta}}\|_2^2 \mathrm{tr}(\mathbf{\Sigma})})$, here $n\|\overline{\boldsymbol{\beta}}\|_2^2 \mathrm{tr}(\mathbf{\Sigma}) \gg 1$. These combined outcomes signify that the variation in the weight vectors $(\mathbf{w}_r(t) - \mathbf{w}_r(0))$ aligns with $\mathbf{X}^\top \mathbf{y}$. Besides, the neural tangent kernel of the neural network adheres to a specific structure, $\widehat{\Theta} = \mathbf{X} \left( \mathbf{I} + \frac{1}{\gamma} \mathbf{X}^\top \mathbf{y} \mathbf{y}^\top \mathbf{X} \right)$,

which diverges from the minimum norm solution. This divergence is attributed to the evolution of $(\mathbf{w}_r(t) - \mathbf{w}_r(0))$, capturing a sufficient extent of the feature vector.

**Phases 2: Interpolating phase**  In this stage, as the output of the neural network increases, terms in the dynamics described by Equations (7) and (8) such as $\left(\frac{\partial f(\mathbf{X})}{\partial \mathbf{w}_r(t)}\right)^\top f(\mathbf{X}, t)$ and $\left(\frac{\partial f(\mathbf{X})}{\partial v_r(t)}\right)^\top f(\mathbf{X}, t)$ become substantial and profoundly influence the optimization trajectory. Consequently, we meticulously assess the precise impact of these terms on the evolution of $\rho_r(t)$ and $v_r(t)$ and show that the $|\rho_r(t)|$ will incrementally increase until the training loss achieve $\epsilon$. Thanks to the analysis in phase 1, we know that for $r \in [m]$, $\rho_r(t_1)$ is significantly larger than $v_r(t_1)$ and the ratio $\frac{\rho_r(t)}{v_r(t)} = \Theta(\sqrt{\gamma})$. By leveraging unchanged ratio between $\rho_r(t)$ and $v_r(t)$ and the monotonicity of $\rho_r(t)v_r(t)$, we can find the solution to the dynamics in the second stage.

It approximately holds that $\mathbf{w}_r(t) \approx \mathbf{w}_r(0) + \rho_r(t)/\gamma \cdot \mathbf{X}^\top \mathbf{y}$ for all $r \in [m]$. Then the feature learning dynamics simplifies to the following expression:

$$\rho_r(t+1) \approx \rho_r(t) + \frac{\eta\gamma}{nm} v_r(t) - \frac{\eta\phi}{nm\gamma} v_r^2(t)\rho_r(t), \quad v_r(t+1) \approx v_r(t) + \frac{\eta}{nm}\rho_r(t) - \frac{\eta\phi}{nm\gamma^2}\rho_r^2(t)v_r(t),$$
(10)

where we define $\phi = \mathbf{y}^\top \mathbf{X}\mathbf{X}^\top \mathbf{X}\mathbf{X}^\top \mathbf{y}$. From Equation (10), we observed that $\rho_r(t)$ and $v_r(t)$ correlate with each other and together they form a two-dimensional nonlinear dynamical system. In the second stage, non-linear terms become significant. Using the initialization conditions from Lemma 5.2, we find that $\rho_r(t) \approx \sqrt{\gamma}v_r(t)$ holds throughout Stage 2. Therefore, plugging $\rho_r(t) = \sqrt{\gamma}v_r(t)$ into Equation (10), we then obtain a unified equation:

$$\rho_r(t+1) = \rho_r(t) + \frac{\eta\sqrt{\gamma}}{nm} \cdot \rho_r(t) - \frac{\eta\phi}{nm\gamma^2} \cdot \rho_r^3(t).$$
(11)

Equation (11) implies that $|\rho_r(t)|$ consistently increases during the second stage, converging towards a value where $\rho_r^2(t) = \gamma^{5/2}/\phi$. However, as $\rho_r^2(t)$ approaches $\gamma^{5/2}/\phi$, the growth rate of $|\rho_r(t)|$ slows down, making the dynamics in Equation (11) challenging to track due to its inherent non-linearity. To characterize the time that $\rho_r(t)$ will spend in the second stage, a proxy $\tilde{\rho}_r(t) \triangleq \sqrt{\frac{\rho_r^2(t)}{1-\phi/\gamma^{5/2}\rho_r^2(t)}}$, is introduced. This proxy is shown to escalate exponentially until it reaches a value of $\sqrt{\frac{(1-\epsilon)\|\mathbf{y}\|_2^2}{n\epsilon\phi/\gamma^5/2}}$. The summarized findings are encapsulated in the following lemma:

**Lemma 5.3.** *Under Assumption 4.1, for any $\epsilon > 0$, there exists a time step $t_2$ satisfying $t_2 - t_1 = \Omega\left(\frac{m\sqrt{n}}{\eta\|\overline{\boldsymbol{\beta}}\|_2\sqrt{\mathrm{tr}(\boldsymbol{\Sigma})}}\log(\frac{n\|\overline{\boldsymbol{\beta}}\|_2\mathrm{tr}(\boldsymbol{\Sigma})}{\epsilon})\right)$ such that $L(f(\mathbf{X}, t_2), \mathbf{y}) \leq 4\epsilon$. Besides, during the second stage $t_1 \leq t \leq t_2$, we have*

- $\|\mathbf{w}_r(t) - \mathbf{w}_r(0) - \rho(t)\mathbf{X}^\top \mathbf{y}\|_2 \leq \frac{1}{2}\rho_r(t)/\sqrt{\mathrm{tr}(\boldsymbol{\Sigma})n\|\overline{\boldsymbol{\beta}}\|_2^2}$.

- $|\rho_r(t) - v_r(t)\sqrt{\gamma}| \leq \frac{1}{2}\rho_r(t)$.

- $\left\|\Theta(\mathbf{X}, \mathbf{X}, t) - \frac{v_r^2(t)}{m}\mathbf{X}\left(\mathbf{I} + \frac{1}{\gamma}\mathbf{X}^\top \mathbf{y}\mathbf{y}^\top \mathbf{X}\right)\mathbf{X}^\top\right\|_2 \leq C\frac{v_r^2(t)\mathrm{tr}(\boldsymbol{\Sigma})}{m}$.

Lemma 5.3 demonstrates the learning trajectory during the second stage. Initially, $\mathbf{w}_r(t)$ aligns with the direction of $\mathbf{X}^\top \mathbf{y}$. As learning progresses, $\mathbf{w}_r(t)$ continuously evolves in the feature vector's direction until the output function approximates the ground truth. Throughout this stage, the magnitude of the neural tangent kernel gradually escalates.

# 6. Experiment

In this section, we experimentally verify our stage-wise analysis of training dynamics. Specifically, we target validating three key behaviors: (i) Training loss should be barely optimized in the first stage, and quickly drop in the second stage. (ii) $\rho_r(t)/v_r(t)$ should increase in the first few steps,

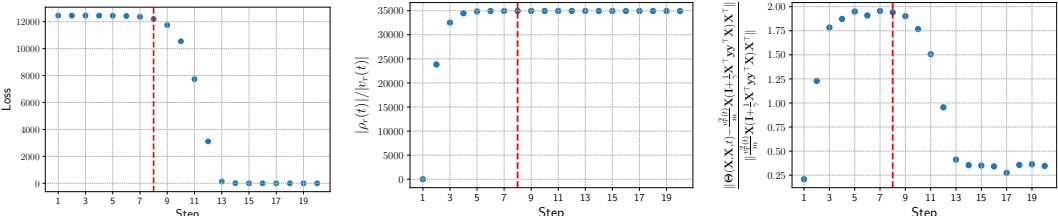

Figure 1: We verify our two-stage training dynamics with synthetic data. The red vertical line indicates empirical $t_1$, separating the first stage (where loss is barely optimized but model weights become aligned with the data) and the second stage (where loss is minimized with stable ratio between $\rho(t)$ and $v(t)$. For the subscript $r$, we average dimensions over the model's hidden size.

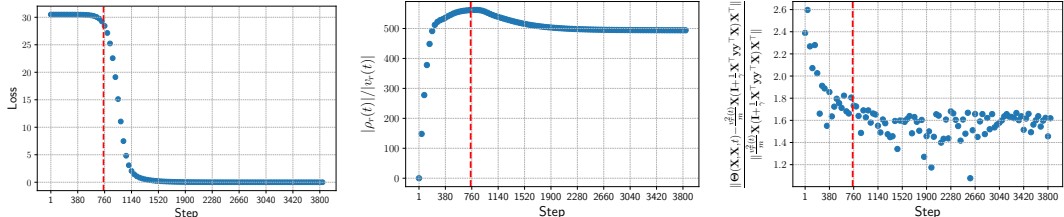

Figure 2: We verify our two-stage training dynamics with CIFAR-10 (with MSE regression loss). The red vertical line indicates empirical $t_1$, separating the first stage (loss remains high but model weights keep aligning with the data), and the second stage (loss drops quickly with stable $\rho(t), v(t)$.

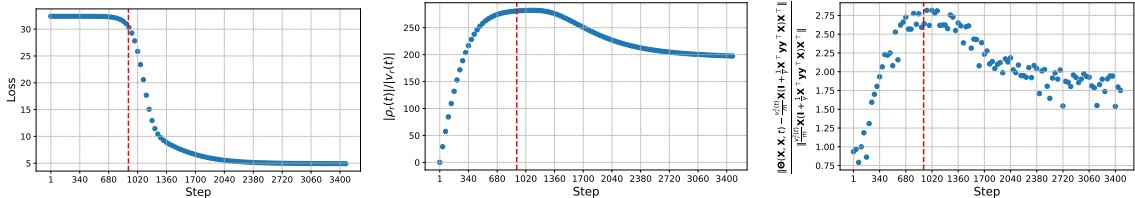

Figure 3: We observe the two-stage training dynamics with CIFAR-10 on a two-layer ReLU network. The red vertical line indicates empirical $t_1$, separating the first stage (loss remains high but model weights keep aligning with the data), and the second stage (loss drops quickly with stable $\rho(t)$ and $v(t)$. For the subscript $r$, we average all dimensions over the model's hidden size.

and remain stable in subsequent steps. (iii) NTK $\mathbf{\Theta}(\mathbf{X}, \mathbf{X}, t)$ converges to the target-aligned kernel $\frac{v_r^2(t)}{m}\mathbf{X}\left(\mathbf{I} + \frac{1}{\gamma}\mathbf{X}^\top \mathbf{y}\mathbf{y}^\top \mathbf{X}\right)\mathbf{X}^\top$ after $t_1$.

**Synthetic Data** We choose our experimental settings as: model's hidden size $m = 1024$, input dimension $d = 10000$, learning rate $\eta = 0.1$, the number of training samples $n = 10$, the variance of the first layer's weight initialization $\sigma_0 = 0.01$, the scaling of the second layer $v_0 = 0.001$, and the variance for the noise term in data $\sigma_y^2 = 0.01$. We sample i.i.d. elements of both $\mathbf{x}$ and $\boldsymbol{\beta}$ from the standard normal distribution.

As shown in Figure 1, we can empirically verify this two-stage training dynamics. During early training steps, the loss remains high and is not actively minimized. However, the model weights quickly align with the data and become stable, as indicated by $\rho_r(t)/v_r(t)$. After this alignment between weights and data, the loss starts being actively minimized and significantly drops. Meanwhile, the weight's alignment $\rho_r(t)/v_r(t)$ remains stable. Moreover, the empirical NTK converges to a target-aligned kernel. The red vertical line marks the empirical $t_1$ that separates the first and the second training stages.

**CIFAR-10** We extended our study of training dynamics to real-world data, as illustrated in Figure 2 for two-layer linear network. Here, we observed similar two-stage training dynamics on the CIFAR-10 dataset, using a regression loss. During the initial training steps, the loss remains considerably

high, while the model weights exhibit substantial alignment with the data. In the second training stage, the loss decreases, and the alignment of the weights, represented as $\rho_r(t)/v_r(t)$, maintains stability. Furthermore, in this stage, the empirical NTK converges towards a kernel that aligns with the target.

**ReLU network** We extended our study of training dynamics to non-linear neural networks, as illustrated in Figure 3, using a two-layer ReLU network. Here, we observed similar two-stage training dynamics on the CIFAR-10 dataset with a regression loss. During the initial training steps, the loss remains considerably high, while the model weights exhibit substantial alignment with the data. In the second training stage, the loss decreases, and the alignment of the weights, represented as $\rho_r(t)/v_r(t)$, remains stable. Furthermore, during this stage, the empirical NTK converges towards a kernel that aligns with the target. These findings suggest that our theoretical analysis may be extendable to non-linear neural networks.

## 7. Conclusion and Discussion

This paper employs non-linear system analysis to investigate the feature learning dynamics in the training of a two-layer linear neural network. Unlike the lazy training process, we explicitly define the conditions under which the neural network will primarily concentrate on feature learning. Consequently, we reveal that the network exhibits a rich two-stage dynamics beyond lazy training in the regression setting. Different from most previous theoretical researches on the two-layer network in the regression setting, this study presents a full and exact training dynamics of gradient descent in the interpolation regime. A significant direction for future work involves studying the feature learning of deep linear neural networks and non-linear networks in a regression setting.

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

# A. Preliminary Lemmas

In this section, we present some pivotal lemmas that give some important properties of the data and the neural network parameters at their random initialization.

**Lemma A.1.** *Denote that $\gamma \triangleq \mathbf{y}^\top \mathbf{X} \mathbf{X}^\top \mathbf{y}$. Suppose that $m = \Omega(\log(1/\delta))$, then with probability at least $1 - \delta$, for all $r \in [m]$,*

$$|\langle \mathbf{w}_r(0), \mathbf{X}^\top \mathbf{y} \rangle| \leq \sigma_0 \sqrt{2 \log(8m/\delta)\gamma},$$
$$\sigma_0 \sqrt{\gamma}/2 \leq \max_{r \in [m]} \langle \mathbf{w}_r(0), \mathbf{X}^\top \mathbf{y} \rangle \leq \sigma_0 \sqrt{2 \log(8m/\delta)\gamma}.$$

*Proof of Lemma A.1.* It is clear that for each $r \in [m]$, $\langle \mathbf{w}_r(0), \mathbf{X}^\top \mathbf{y} \rangle$ is a Gaussian random variable with mean zero and variance $\sigma_0^2 \gamma$. Therefore, by Gaussian tail bound and union bound, with probability at least $1 - \delta$,

$$|\langle \mathbf{w}_r(0), \mathbf{X}^\top \mathbf{y} \rangle| \leq \sqrt{2 \log(8m/\delta)} \cdot \sigma_0 \|\mathbf{X}^\top \mathbf{y}\|_2.$$

Moreover, $P(\sigma_0 \sqrt{\gamma}/2 > \langle \mathbf{w}_r(0), \mathbf{X}^\top \mathbf{y} \rangle)$ is an absolute constant, and therefore by the condition on $m$, we have

$$P\left(\sigma_0 \sqrt{\gamma}/2 \leq \max_{r \in [m]} \langle \mathbf{w}_r(0), \mathbf{X}^\top \mathbf{y} \rangle\right) = 1 - P(\sigma_0 \sqrt{\gamma}/2 > \max_{r \in [m]} j \cdot \langle \mathbf{w}_r(0), \mathbf{X}^\top \mathbf{y} \rangle)$$
$$= 1 - P(\sigma_0 \sqrt{\gamma}/2 > \langle \mathbf{w}_r(0), \mathbf{X}^\top \mathbf{y} \rangle)^{2m}$$
$$\geq 1 - \delta/4.$$

$\square$

**Lemma A.2.** *Suppose that $\delta > 0$ and $d = \Omega(\log(4m/\delta))$. Then with probability at least $1 - \delta$, for all $r \in [m]$,*

$$\sigma_0^2 d/2 \leq \|\mathbf{w}_r(0)\|_2^2 \leq 3\sigma_0^2 d/2.$$

*Proof of Lemma A.2.* By Bernstein's inequality, with probability at least $1 - \delta/(2m)$ we have

$$\left| \|\mathbf{w}_r(0)\|_2^2 - \sigma_0^2 d \right| = O(\sigma_0^2 \cdot \sqrt{d \log(4m/\delta)}).$$

Therefore, as long as $d = \Omega(\log(4m/\delta))$, we have

$$\sigma_0^2 d/2 \leq \|\mathbf{w}_r(0)\|_2^2 \leq 3\sigma_0^2 d/2.$$

$\square$

**Lemma A.3.** *Suppose that $\mathbf{Z} \in \mathbb{R}^{n \times d}$ is a random matrix with i.i.d. sub-Gaussian entries with sub-Gaussian norm $\sigma_x$ and data is generated through $\mathbf{X} = \mathbf{Z}\mathbf{\Lambda}^{\frac{1}{2}}\mathbf{V}^\top$. Given a vector $\boldsymbol{\alpha} \in \mathbb{R}^d$. Then with probability at least $1 - O(n^{-2})$, we have,*

$$n/2\|\boldsymbol{\alpha}\|_{\boldsymbol{\Sigma}}^2 \leq \boldsymbol{\alpha}^\top \mathbf{X}^\top \mathbf{X} \boldsymbol{\alpha} \leq 2n\|\boldsymbol{\alpha}\|_{\boldsymbol{\Sigma}}^2.$$

*Proof of Lemma A.3.* We expand the expression by the definition of $\mathbf{X}$:

$$\boldsymbol{\alpha}^\top \mathbf{X}^\top \mathbf{X} \boldsymbol{\alpha} = \boldsymbol{\alpha}^\top \mathbf{V} \mathbf{\Lambda}^{\frac{1}{2}} \mathbf{Z}^\top \mathbf{Z} \mathbf{\Lambda}^{\frac{1}{2}} \mathbf{V}^\top \boldsymbol{\alpha}$$
$$= \widetilde{\boldsymbol{\alpha}}^\top \mathbf{Z}^\top \mathbf{Z} \widetilde{\boldsymbol{\alpha}}$$
$$= \sum_{i=1}^n \left( \sum_{j=1}^d \widetilde{\alpha}_j Z_{ij} \right)^2,$$

where we define $\widetilde{\mathbf{a}} = \mathbf{\Lambda}^{\frac{1}{2}}\mathbf{V}^\top\boldsymbol{\alpha}$. Besides, $Z_{ij}$ is an entry of the matrix $\mathbf{Z}$ and $\widetilde{a}_j$ is an element of the vector $\widetilde{\boldsymbol{\alpha}}_j$. Note that $Z_{ij}$ is for $i \in [n]$ and $j \in [d]$ are independent sub-Gaussian vectors with $\|Z_{ij}\|_{\psi_2} < \sigma_x$, Then by Lemma 5.9 in [77],

$$\left\|\sum_{j=1}^d \widetilde{\alpha}_j Z_{ij}\right\|_{\psi_2} \le c_1\|\widetilde{\boldsymbol{\alpha}}\|_2 \cdot \sigma_x,$$

where $c_1$ is an absolute constant. Therefore by Lemma 5.14 in [77], we have

$$\left\|\left(\sum_{j=1}^d \widetilde{\alpha}_j Z_{ij}\right)^2 - \|\widetilde{\boldsymbol{\alpha}}\|_2^2\right\|_{\psi_1} \le c_2\|\widetilde{\boldsymbol{\alpha}}\|_2^2 \cdot \sigma_x,$$

where $c_2$ is an absolute constant and we merge $\sigma_x$ into the constant $c_2$. By Bernstein's inequality, with probability at least $1 - \delta$ we have,

$$\left|\|\mathbf{Z}\widetilde{\boldsymbol{\alpha}}\|_2^2 - \mathbb{E}\big[\|\mathbf{Z}\widetilde{\boldsymbol{\alpha}}\|_2^2\big]\right| \le c_3\|\widetilde{\boldsymbol{\alpha}}\|_2^2\sqrt{\log(1/\delta)n/2},$$

where $c_3$ is an absolute constant. Therefore with probability at least $1 - n^{-2}$ we have

$$n\|\widetilde{\boldsymbol{\alpha}}\|_2^2 - c_3\|\widetilde{\boldsymbol{\alpha}}\|_2^2\sqrt{\log(n)n} \le \|\mathbf{Z}\widetilde{\boldsymbol{\alpha}}\|_2^2 \le n\|\widetilde{\boldsymbol{\alpha}}\|_2^2 + c_3\|\widetilde{\boldsymbol{\alpha}}\|_2^2\sqrt{\log(n)n}.$$

Therefore we can further obtain the desired result:

$$n/2\|\widetilde{\boldsymbol{\alpha}}\|_2^2 \le \|\mathbf{Z}\widetilde{\boldsymbol{\alpha}}\|_2^2 \le 2n\|\widetilde{\boldsymbol{\alpha}}\|_2^2,$$

where $\|\widetilde{\boldsymbol{\alpha}}\|_2 = \|\boldsymbol{\alpha}\|_{\mathbf{\Sigma}}$.

$\square$

**Lemma A.4.** *Suppose that* $\mathbf{X} = \mathbf{Z}\mathbf{\Lambda}^{\frac{1}{2}}\mathbf{V}^\top$ *where* $\mathbf{Z} \in \mathbb{R}^{n \times d}$ *is a random matrix with i.i.d. sub-Gaussian entries with sub-Gaussian norm* $\sigma_x$*. Given that* $\sigma_y^2 \le 8n\|\overline{\boldsymbol{\beta}}\|_2^2$*, with probability at least* $1 - O(n^{-2})$ *it holds that*

$$\frac{1}{2}\mathrm{tr}(\mathbf{\Sigma})n\|\overline{\boldsymbol{\beta}}\|_2^2 \le \mathbf{y}^\top\mathbf{X}\mathbf{X}^\top\mathbf{y} \le \frac{3}{2}\mathrm{tr}(\mathbf{\Sigma})n\|\overline{\boldsymbol{\beta}}\|_2^2.$$

*Proof.* Working at the upper bound, we first have

$$\begin{aligned}
\mathbf{y}^\top\mathbf{X}\mathbf{X}^\top\mathbf{y} &= \mathbf{y}^\top(\mathbf{Z}\mathbf{\Lambda}\mathbf{Z}^\top)\mathbf{y} \\
&\le \|\mathbf{y}\|_2^2 \cdot (\mathrm{tr}(\mathbf{\Sigma}) + \epsilon_\lambda) \\
&= \|\mathbf{y}\|_2^2 \cdot (\mathrm{tr}(\mathbf{\Sigma}) + C\sigma_x^2(n \cdot \|\mathbf{\Sigma}\|_2 + \sqrt{n} \cdot \|\mathbf{\Sigma}\|_F)).
\end{aligned}$$

The first inequity is by Lemma 5.1. Similarly, by Lemma 5.1, we obtain the lower bound

$$\begin{aligned}
\mathbf{y}^\top\mathbf{X}\mathbf{X}^\top\mathbf{y} &= \mathbf{y}^\top(\mathbf{Z}\mathbf{\Lambda}\mathbf{Z}^\top)\mathbf{y} \\
&\ge \|\mathbf{y}\|_2^2 \cdot (\mathrm{tr}(\mathbf{\Sigma}) - \epsilon_\lambda) \\
&= \|\mathbf{y}\|_2^2 \cdot (\mathrm{tr}(\mathbf{\Sigma}) - C\sigma_x^2(n \cdot \|\mathbf{\Sigma}\|_2 + \sqrt{n} \cdot \|\mathbf{\Sigma}\|_F)) \\
&\ge \frac{1}{2}\mathrm{tr}(\mathbf{\Sigma})\|\mathbf{y}\|_2^2,
\end{aligned}$$

where the last inequality is by the condition that $\mathrm{tr}(\mathbf{\Sigma}) \ge 2C_3\sigma_x^2(n \cdot \|\mathbf{\Sigma}\|_2 + \sqrt{n} \cdot \|\mathbf{\Sigma}\|_F)$. Next, we compute the upper bound of $\|\mathbf{y}\|_2^2$ as follows

$$\begin{aligned}
\mathbf{y}^\top\mathbf{y} &= (\mathbf{X}\boldsymbol{\beta} + \boldsymbol{\epsilon})^\top(\mathbf{X}\boldsymbol{\beta} + \boldsymbol{\epsilon}) \\
&\le 2n\|\widetilde{\boldsymbol{\beta}}\|_2^2 + 2\boldsymbol{\epsilon}^\top\mathbf{X}\boldsymbol{\beta} + \boldsymbol{\epsilon}^\top\boldsymbol{\epsilon} \\
&\le \frac{3}{2}n\|\widetilde{\boldsymbol{\beta}}\|_2^2.
\end{aligned}$$

The first inequality is by Lemma A.3, and the second inequality is by $n \geq \frac{4\sigma_y^2}{\|\widetilde{\boldsymbol{\beta}}\|_2^2}$. Similarly, we obtain the lower bound

$$
\begin{aligned}
\mathbf{y}^\top \mathbf{y} &= (\mathbf{X}\boldsymbol{\beta} + \boldsymbol{\epsilon})^\top (\mathbf{X}\boldsymbol{\beta} + \boldsymbol{\epsilon}) \\
&\geq \frac{1}{2}n\|\widetilde{\boldsymbol{\beta}}\|_2^2 + 2\boldsymbol{\epsilon}^\top \mathbf{X}\boldsymbol{\beta} + \boldsymbol{\epsilon}^\top \boldsymbol{\epsilon} \\
&\geq \frac{1}{4}n\|\widetilde{\boldsymbol{\beta}}\|_2^2.
\end{aligned}
$$

The first inequality is by Lemma A.3, and the second inequality is by $\sigma_y^2 \leq 8n\|\widetilde{\boldsymbol{\beta}}\|_2^2$. Together, we conclude that

$$
\begin{aligned}
\mathbf{y}^\top \mathbf{X}\mathbf{X}^\top \mathbf{y} &= \mathbf{y}^\top (\mathbf{Z}\boldsymbol{\Lambda}\mathbf{Z}^\top)\mathbf{y} \\
&\leq \|\mathbf{y}\|_2^2 \cdot (\operatorname{tr}(\boldsymbol{\Sigma}) + \epsilon_\lambda) \\
&= \|\mathbf{y}\|_2^2 \cdot (\operatorname{tr}(\boldsymbol{\Sigma}) + C\sigma_x^2(n \cdot \|\boldsymbol{\Sigma}\|_2 + \sqrt{n} \cdot \|\boldsymbol{\Sigma}\|_F)) \\
&\leq \frac{3}{2}\|\mathbf{y}\|_2^2 \operatorname{tr}(\boldsymbol{\Sigma}).
\end{aligned}
$$

$\square$

# B. Stage-wise analysis for gradient descent dynamics

By the definition of feature alignment $\rho_r(t) = \langle \mathbf{w}_r(t) - \mathbf{w}_r(0), \mathbf{X}^\top \mathbf{y} \rangle$, the gradient descent update

$$
\begin{aligned}
\mathbf{w}_r(t+1) &= \mathbf{w}_r(t) + \frac{\eta}{mn} \cdot v_r(t) \cdot \mathbf{X}^\top \mathbf{y} - \frac{\eta}{m^2n} \cdot v_r(t) \cdot \mathbf{X}^\top \mathbf{X}\mathbf{W}(t)\mathbf{v}(t), \\
v_r(t+1) &= v_r(t) + \frac{\eta}{mn} \cdot \mathbf{w}_r^\top(t)\mathbf{X}^\top \mathbf{y} - \frac{\eta}{m^2n} \cdot \mathbf{w}_r^\top(t)\mathbf{X}^\top \mathbf{X}\mathbf{W}(t)\mathbf{v}(t).
\end{aligned}
\tag{12}
$$

will result in:

$$
\begin{aligned}
\rho_r(t+1) &= \rho_r(t) + \frac{\eta}{mn} \cdot v_r(t) \cdot \langle \mathbf{X}^\top \mathbf{y}, \mathbf{X}^\top \mathbf{y} \rangle - \frac{\eta}{m^2n} \cdot v_r(t) \cdot \langle \mathbf{X}^\top \mathbf{X}\mathbf{W}(t)\mathbf{v}(t), \mathbf{X}^\top \mathbf{y} \rangle, \\
v_r(t+1) &= v_r(t) + \frac{\eta}{mn} \cdot \rho_r(t) + \frac{\eta}{mn} \cdot \langle \mathbf{w}_r(0), \mathbf{X}^\top \mathbf{y} \rangle - \frac{\eta}{m^2n} \cdot \langle \mathbf{w}_r(t), \mathbf{X}^\top \mathbf{X}\mathbf{W}(t)\mathbf{v}(t) \rangle.
\end{aligned}
$$

In the first stage, when we bound the last gradient descent term in Equation (2) and $\langle \mathbf{w}_r(0), \mathbf{X}^\top \mathbf{y} \rangle$, then the non-linear dynamics system nearly reduce to a linear system. By this, we characterize the behavior of neural network.

**Lemma B.1** (Restatement of Lemma 5.2). *Under assumption 4.1, there exists a time step* $t_1 = \frac{\log(1/v_0)m\sqrt{n}}{\eta\sqrt{\operatorname{tr}(\boldsymbol{\Sigma})}\|\overline{\boldsymbol{\beta}}\|_2}$, *with probability at least* $1 - O(n^{-2})$, *such that*

- *for all* $r \in [m]$, *we have* $\rho_r(t_1) = \Theta(1)$.

- *for all* $r \in [m]$, *we have* $v_r(t_1) = \Theta(1/\sqrt{\operatorname{tr}(\boldsymbol{\Sigma})n\|\overline{\boldsymbol{\beta}}\|_2^2})$.

- $\left\| \Theta(\mathbf{X}, \mathbf{X}, t_1) - \frac{1}{m\gamma}\mathbf{X}\left(\mathbf{I} + \frac{1}{\gamma}\mathbf{X}^\top \mathbf{y}\mathbf{y}^\top \mathbf{X}\right)\mathbf{X}^\top \right\|_2 \leq \frac{C}{mn^3\operatorname{tr}(\boldsymbol{\Sigma})\|\overline{\boldsymbol{\beta}}\|_2^6}$.

*where $C$ is an absolute constant.*

*Proof of Lemma B.1.* According to the gradient descent update rule, we know the evolution equation for $\rho_r(t)$ and $v_r(t)$ follows:

$$
\begin{aligned}
\rho_r(t+1) &= \rho_r(t) + \frac{\eta}{mn} \cdot v_r(t) \cdot \gamma - \frac{\eta}{m^2n} \cdot v_r(t) \cdot \langle \mathbf{X}^\top \mathbf{X}\mathbf{W}(t)\mathbf{v}(t), \mathbf{X}^\top \mathbf{y} \rangle, \\
v_r(t+1) &= v_r(t) + \frac{\eta}{mn} \cdot \rho_r(t) + \frac{\eta}{mn} \cdot \langle \mathbf{w}_r(0), \mathbf{X}^\top \mathbf{y} \rangle - \frac{\eta}{m^2n} \cdot \langle \mathbf{w}_r(t), \mathbf{X}^\top \mathbf{X}\mathbf{W}(t)\mathbf{v}(t) \rangle.
\end{aligned}
$$

First, we adopt an induction method to show during the first stage $0 \leq t \leq t_1$, where $t_1 = \frac{C \log(1/v_0)mn}{\eta\sqrt{\text{tr}(\mathbf{\Sigma})n}\|\overline{\boldsymbol{\beta}}\|_2^2}$, with probabilities at least $1 - n^{-2}$, the following inequalities hold for all $r \in [m]$:

$$\rho_r(t)v_r(0) \leq \left(1 + \frac{\eta\sqrt{\gamma_u}}{nm}\right)^t \left(\frac{\sqrt{\gamma}v_0^2}{2}\right) - \left(1 - \frac{\eta\sqrt{\gamma_u}}{nm}\right)^t \left(\frac{\sqrt{\gamma}v_0^2}{2}\right).$$

$$\rho_r(t)v_r(0) \geq \left(1 + \frac{\eta\sqrt{\gamma_l}}{nm}\right)^t \left(\frac{\sqrt{\gamma}v_0^2}{2}\right) - \left(1 - \frac{\eta\sqrt{\gamma_l}}{nm}\right)^t \left(\frac{\sqrt{\gamma}v_0^2}{2}\right).$$

$$v_r(t)v_r(0) \leq \left(1 + \frac{\eta\sqrt{\gamma_u}}{nm}\right)^t \left(\frac{v_0^2}{2}\right) + \left(1 - \frac{\eta\sqrt{\gamma_u}}{nm}\right)^t \left(\frac{v_0^2}{2}\right). \tag{13}$$

$$v_r(t)v_r(0) \geq \left(1 + \frac{\eta\sqrt{\gamma_l}}{nm}\right)^t \left(\frac{v_0^2}{2}\right) + \left(1 - \frac{\eta\sqrt{\gamma_l}}{nm}\right)^t \left(\frac{v_0^2}{2}\right).$$

$$\|\mathbf{w}_r(t) - \mathbf{w}_r(0) - \rho_r(t)/\gamma \cdot \mathbf{X}^\top \mathbf{y}\|_2 \leq C\frac{\text{tr}(\mathbf{\Sigma})}{\gamma^{3/2}}\rho_r(t)/\sqrt{\gamma}.$$

where $\gamma_l = \gamma\left(1 - C_1\frac{\text{tr}(\mathbf{\Sigma})}{\gamma^{3/2}}\right)^2$ and $\gamma_u = \gamma\left(1 + C_2\frac{\text{tr}(\mathbf{\Sigma})}{\gamma^{3/2}}\right)^2$, with $C_1$ and $C_2$ being absolute constant.

It is straightforward to see that at $t = 0$ the above inequalities are all satisfied. Given that for $t \leq t_1$, we assume all inequalities hold for $t$.

**First claim**. Denote that $\epsilon_\lambda = C\sigma_x^2(n \cdot \|\mathbf{\Sigma}\|_2 + \sqrt{n} \cdot \|\mathbf{\Sigma}\|_F)$. Then we proceed to the time step of $t + 1$ for the first induction claim in Equation (13):

$$\rho_r(t+1)v_r(0) = \rho_r(t)v_r(0) + \frac{\eta}{mn} \cdot v_r(t)v_r(0) \cdot \gamma - \frac{\eta}{m^2n} \cdot v_r(t)v_r(0) \cdot \langle \mathbf{X}^\top \mathbf{X}\mathbf{W}(t)\mathbf{v}(t), \mathbf{X}^\top \mathbf{y}\rangle$$

$$= \rho_r(t)v_r(0) + \frac{\eta}{mn} \cdot v_r(t)v_r(0) \cdot \gamma\left(1 - \frac{1}{m\gamma} \cdot \mathbf{y}^\top \mathbf{X}\mathbf{X}^\top \mathbf{X}\mathbf{W}(t)\mathbf{v}(t)\right)$$

$$\geq \rho_r(t)v_r(0) + \frac{\eta}{mn} \cdot v_r(t)v_r(0) \cdot \gamma\left(1 - \frac{\text{tr}(\mathbf{\Sigma}) + \epsilon_\lambda}{m\gamma} \cdot \mathbf{y}^\top \mathbf{X}\mathbf{W}(t)\mathbf{v}(t)\right)$$

$$= \rho_r(t)v_r(0) + \frac{\eta}{mn} \cdot v_r(t)v_r(0) \cdot \gamma\left(1 - \frac{\text{tr}(\mathbf{\Sigma}) + \epsilon_\lambda}{m\gamma} \cdot \sum_{r'=1}^m \mathbf{y}^\top \mathbf{X}\mathbf{w}_{r'}(t)v_{r'}(t)\right)$$

$$= \rho_r(t)v_r(0) + \frac{\eta}{mn} \cdot v_r(t)v_r(0) \cdot \gamma\left(1 - \frac{\text{tr}(\mathbf{\Sigma}) + \epsilon_\lambda}{m\gamma}\right.$$

$$\left.\left(\sum_{r'=1}^m \rho_{r'}(t)v_{r'}(t) + \sum_{r'=1}^m \langle \mathbf{w}_{r'}(0), \mathbf{X}^\top \mathbf{y}\rangle v_{r'}(t)\right)\right)$$

$$\geq \rho_r(t)v_r(0) + \frac{\eta}{mn}v_r(t)v_r(0)\gamma\left(1 - \frac{\text{tr}(\mathbf{\Sigma}) + \epsilon_\lambda}{m\gamma}\left(\sum_{r'=1}^m \rho_{r'}(t)v_{r'}(t) + \sum_{r'=1}^m 0.1v_{r'}(t)\right)\right)$$

$$\geq \rho_r(t)v_r(0) + \frac{\eta}{mn} \cdot v_r(t)v_r(0) \cdot \gamma\left(1 - 1.1\frac{\text{tr}(\mathbf{\Sigma}) + \epsilon_\lambda}{\gamma^{3/2}}\right).$$

The second inequality is by Lemma A.1 and Assumption 4.1 on $\sigma_0$. Finally, the last inequality is by the fact that when $t \leq t_1$, for all $r \in [m]$, $\rho_r(t) \leq 1$ and $v_r(t) \leq 1/\sqrt{\gamma}$. Additionally, we apply the

induction assumption to $v_r(t)$ and $\rho_r(t)$, yielding the following:

$$
\begin{aligned}
&\rho_r(t+1)v_r(0)\\
&\geq \rho_r(t)v_r(0) + \frac{\eta}{mn}\cdot v_r(t)v_r(0)\cdot\sqrt{\gamma}\sqrt{\gamma_l}\\
&\geq \left(1+\frac{\eta\sqrt{\gamma_l}}{nm}\right)^t\frac{v_r^2(0)\sqrt{\gamma}}{2} - \left(1-\frac{\eta\sqrt{\gamma_l}}{nm}\right)^t\frac{v_r^2(0)\sqrt{\gamma}}{2} + \frac{\eta\sqrt{\gamma_l}}{nm}\left(1+\frac{\eta\sqrt{\gamma_l}}{nm}\right)^t\frac{v_r^2(0)\sqrt{\gamma}}{2}\\
&\quad + \frac{\eta\sqrt{\gamma_l}}{nm}\left(1-\frac{\eta\sqrt{\gamma_l}}{nm}\right)^t\frac{v_r^2(0)\sqrt{\gamma}}{2}\\
&= \left(1+\frac{\eta\sqrt{\gamma_l}}{nm}\right)^{t+1}\frac{v_r^2(0)\sqrt{\gamma}}{2} - \left(1-\frac{\eta\sqrt{\gamma_l}}{nm}\right)^{t+1}\frac{v_r^2(0)\sqrt{\gamma}}{2}.
\end{aligned}
$$

This concludes the proof of the first induction claim.

**Second claim**. The next step is to prove the second induction claim:

$$
\begin{aligned}
&\rho_r(t+1)v_r(0)\\
&= \rho_r(t)v_r(0) + \frac{\eta}{mn}\cdot v_r(t)v_r(0)\cdot\gamma - \frac{\eta}{m^2n}\cdot v_r(t)v_r(0)\cdot\langle\mathbf{X}^\top\mathbf{X}\mathbf{W}(t)\mathbf{v}(t),\mathbf{X}^\top\mathbf{y}\rangle\\
&= \rho_r(t)v_r(0) + \frac{\eta}{mn}\cdot v_r(t)v_r(0)\cdot\gamma\left(1 - \frac{1}{m\gamma}\cdot\mathbf{y}^\top\mathbf{X}\mathbf{X}^\top\mathbf{X}\mathbf{W}(t)\mathbf{v}(t)\right)\\
&\leq \rho_r(t)v_r(0) + \frac{\eta}{mn}v_r(t)v_r(0)\gamma\left(1 - \frac{\mathbf{y}^\top\mathbf{X}\mathbf{X}^\top\mathbf{X}}{m\gamma}(\mathbf{W}(t)-\mathbf{W}(0))\mathbf{v}(t) + \frac{1}{m\gamma}|\mathbf{y}^\top\mathbf{X}\mathbf{X}^\top\mathbf{X}\mathbf{W}(0)\mathbf{v}(t)|\right)\\
&\leq \rho_r(t)v_r(0) + \frac{\eta}{mn}\cdot v_r(t)v_r(0)\cdot\gamma\Bigg(1 - \frac{\text{tr}(\boldsymbol{\Sigma})-\epsilon_\lambda}{m\gamma}\cdot\sum_{r'=1}^m\mathbf{y}^\top\mathbf{X}(\mathbf{w}_{r'}(t)-\mathbf{w}_{r'}(0))v_{r'}(t)\\
&\quad + \frac{\text{tr}(\boldsymbol{\Sigma})+\epsilon_\lambda}{m\gamma}\left|\sum_{r'=1}^m\langle\mathbf{w}_{r'}(0),\mathbf{X}^\top\mathbf{y}\rangle v_{r'}(t)\right|\Bigg)\\
&\leq \rho_r(t)v_r(0) + \frac{\eta}{mn}v_r(t)v_r(0)\gamma\Bigg(1 - \frac{\text{tr}(\boldsymbol{\Sigma})-\epsilon_\lambda}{m\gamma}\sum_{r'=1}^m\rho_{r'}(1)v_{r'}(1)\\
&\quad + \frac{\text{tr}(\boldsymbol{\Sigma})+\epsilon_\lambda}{m\gamma}\sum_{r'=1}^m|\langle\mathbf{w}_{r'}(0),\mathbf{X}^\top\mathbf{y}\rangle|v_{r'}(t)\Bigg)\\
&\leq \rho_r(t)v_r(0) + \frac{\eta}{mn}\cdot v_r(t)v_r(0)\cdot\gamma\left(1 + \frac{\text{tr}(\boldsymbol{\Sigma})+\epsilon_\lambda}{m\gamma}\cdot\sum_{r'=1}^m 0.1 v_{r'}(t)\right)\\
&\leq \rho_r(t)v_r(0) - \frac{\eta}{mn}\cdot v_r(t)v_r(0)\cdot\gamma\left(1 + 0.1\frac{\text{tr}(\boldsymbol{\Sigma})+\epsilon_\lambda}{\gamma^{3/2}}\right).
\end{aligned}
$$

The first inequality arises due to the possibility that $\mathbf{y}^\top\mathbf{X}\mathbf{X}^\top\mathbf{X}\mathbf{W}(0)\mathbf{v}(t)$ can be negative. The third is due to that $\rho_r(t)v_r(t)\geq\rho_r(1)v_r(1)$ for $t\geq 1$ and all $r\in[m]$. Besides, we also apply triangle inequality in the third inequality. The fourth inequality is by the assumption that $\text{tr}(\boldsymbol{\Sigma})>\epsilon_\lambda$, $\rho_r(1)v_r(1)>0$ for all $r\in[m]$, Lemma A.1 and Assumption 4.1 on $\sigma_0$. Finally, the last inequality is by the fact that when $t\leq t_1$, for all $r\in[m]$, $v_r(t)\leq 1/\sqrt{\gamma}$. By further applying the induction assumption to $v_r(t)$

and $\rho_r(t)$, we can conclude the proof of the second claim:

$$\rho_r(t+1)v_r(0)$$
$$\leq \rho_r(t)v_r(0) + \frac{\eta}{mn} \cdot v_r(t)v_r(0) \cdot \sqrt{\gamma}\sqrt{\gamma_l}$$
$$\leq \left(1 + \frac{\eta\sqrt{\gamma_u}}{nm}\right)^t \frac{v_r^2(0)\sqrt{\gamma}}{2} - \left(1 - \frac{\eta\sqrt{\gamma_u}}{nm}\right)^t \frac{v_r^2(0)\sqrt{\gamma}}{2} + \frac{\eta\sqrt{\gamma_u}}{nm}\left(1 + \frac{\eta\sqrt{\gamma_u}}{nm}\right)^t \frac{v_r^2(0)\sqrt{\gamma}}{2}$$
$$+ \frac{\eta\sqrt{\gamma_u}}{nm}\left(1 - \frac{\eta\sqrt{\gamma_u}}{nm}\right)^t \frac{v_r^2(0)\sqrt{\gamma}}{2}$$
$$= \left(1 + \frac{\eta\sqrt{\gamma_u}}{nm}\right)^{t+1} \frac{v_r^2(0)\sqrt{\gamma}}{2} - \left(1 - \frac{\eta\sqrt{\gamma_u}}{nm}\right)^{t+1} \frac{v_r^2(0)\sqrt{\gamma}}{2}.$$

**Third claim**. The next step is to prove the third induction claim for $v_r(t+1)$:

$$v_r(t+1)v_r(0)$$
$$= v_r(t)v_r(0) + \frac{\eta}{mn}\rho_r(t)v_r(0) - \frac{\eta}{m^2n}v_r(0)\mathbf{w}_r^\top(t)\mathbf{X}^\top\mathbf{X}\mathbf{W}(t)\mathbf{v}(t) + \frac{\eta}{mn}\langle\mathbf{w}_r(0), \mathbf{X}^\top\mathbf{y}\rangle v_r(0)$$
$$= v_r(t)v_r(0) + \frac{\eta}{mn}\rho_r(t)v_r(0)\left(1 - \frac{1}{m\rho_r(t)}\mathbf{w}_r^\top(t)\mathbf{X}^\top\mathbf{X}\mathbf{W}(t)\mathbf{v}(t)\right) + \frac{\eta}{mn}\langle\mathbf{w}_r(0), \mathbf{X}^\top\mathbf{y}\rangle v_r(0)$$
$$= v_r(t)v_r(0) + \frac{\eta}{mn}\rho_r(t)v_r(0)\left(1 - \frac{1}{m\rho_r(t)}\sum_{r'=1}^{m}\mathbf{w}_r^\top(t)\mathbf{X}^\top\mathbf{X}\mathbf{w}_{r'}(t)v_{r'}(t)\right) + \frac{\eta\langle\mathbf{w}_r(0), \mathbf{X}^\top\mathbf{y}\rangle v_r(0)}{mn}$$
$$\geq v_r(t)v_r(0) + \frac{\eta}{mn}\rho_r(t)v_r(0)\left(1 - \frac{\mathrm{tr}(\mathbf{\Sigma}) + \epsilon_\lambda}{m\rho_r(t)}\sum_{r'=1}^{m}\mathbf{w}_r^\top(t)\mathbf{w}_{r'}(t)v_{r'}(t)\right) + \frac{\eta}{mn}\langle\mathbf{w}_r(0), \mathbf{X}^\top\mathbf{y}\rangle v_r(0)$$
$$\geq v_r(t)v_r(0) + \frac{\eta\rho_r(t)v_r(0)}{mn}\left(1 - \frac{\mathrm{tr}(\mathbf{\Sigma}) + \epsilon_\lambda}{4m\rho_r(t)}\sum_{r'=1}^{m}\|\mathbf{w}_r(t) + \mathbf{w}_{r'}(t)\|_2^2 v_{r'}(t)\right) + \frac{\eta\langle\mathbf{w}_r(0), \mathbf{X}^\top\mathbf{y}\rangle v_r(0)}{mn}$$
$$\geq v_r(t)v_r(0) + \frac{\eta\rho_r(t)v_r(0)}{mn}\left(1 - \frac{\mathrm{tr}(\mathbf{\Sigma}) + \epsilon_\lambda}{\rho_r(t)\sqrt{\gamma}}(1.01\rho_r(t)/\sqrt{\gamma} + \|\mathbf{w}_r(0)\|_2)^2\right) + \frac{\eta\langle\mathbf{w}_r(0), \mathbf{X}^\top\mathbf{y}\rangle v_r(0)}{mn}$$
$$\geq v_r(t)v_r(0) + \frac{\eta\rho_r(t)v_r(0)}{mn}\left(1 - \frac{\mathrm{tr}(\mathbf{\Sigma}) + \epsilon_\lambda}{\sqrt{\gamma}\rho_r(t)}(1.02\rho_r(t)/\sqrt{\gamma})^2\right) + \frac{\eta\langle\mathbf{w}_r(0), \mathbf{X}^\top\mathbf{y}\rangle v_r(0)}{mn}$$
$$\geq v_r(t)v_r(0) + \frac{\eta}{mn} \cdot \rho_r(t)v_r(0)\left(1 - 1.1\frac{\mathrm{tr}(\mathbf{\Sigma}) + \epsilon_\lambda}{\gamma^{3/2}}\right).$$

The second inequality is by Cauchy–Schwarz inequality and the polarization identity $\boldsymbol{\alpha}^\top\mathbf{A}\boldsymbol{\beta} = 1/4(\boldsymbol{\alpha}+\boldsymbol{\beta})^\top\mathbf{A}(\boldsymbol{\alpha}+\boldsymbol{\beta}) - 1/4(\boldsymbol{\alpha}-\boldsymbol{\beta})^\top\mathbf{A}(\boldsymbol{\alpha}-\boldsymbol{\beta})$. The third inequality is by the fifth induction claim. The fourth inequality is by Lemma A.2 and $\sigma_0 < 0.01/\sqrt{\gamma d}$. Finally, the last inequality is by Lemma A.1. We further take the induction assumption on $v_r(t)$ and $\rho_r(t)$ can conclude the proof on the third claim:

$$v_r(t+1)v_r(0)$$
$$\geq \left(1 + \frac{\eta\sqrt{\gamma_l}}{nm}\right)^t \frac{v_r^2(0)\sqrt{\gamma}}{2} - \left(1 - \frac{\eta\sqrt{\gamma_l}}{nm}\right)^t \frac{v_r^2(0)\sqrt{\gamma}}{2} + \frac{\eta\sqrt{\gamma_l}}{nm}\left(1 + \frac{\eta\sqrt{\gamma_l}}{nm}\right)^t \frac{v_r^2(0)\sqrt{\gamma}}{2}$$
$$+ \frac{\eta\sqrt{\gamma_l}}{nm}\left(1 - \frac{\eta\sqrt{\gamma_l}}{nm}\right)^t \frac{v_r^2(0)\sqrt{\gamma}}{2}$$
$$= \left(1 + \frac{\eta\sqrt{\gamma_l}}{nm}\right)^{t+1} \frac{v_r^2(0)\sqrt{\gamma}}{2} - \left(1 - \frac{\eta\sqrt{\gamma_l}}{nm}\right)^{t+1} \frac{v_r^2(0)\sqrt{\gamma}}{2}.$$

**Fourth claim**. We then provide proof to the fourth induction claim for $v_r(t+1)$:

$$v_r(t+1)v_r(0)$$

$$= v_r(t)v_r(0) + \frac{\eta}{mn} \cdot \rho_r(t)v_r(0) - \frac{\eta}{m^2 n}v_r(0) \cdot \mathbf{w}_r^\top(t)\mathbf{X}^\top \mathbf{X}\mathbf{W}(t)\mathbf{v}(t) + \frac{\eta}{mn}\langle \mathbf{w}_r(0), \mathbf{X}^\top \mathbf{y}\rangle v_r(0)$$

$$= v_r(t)v_r(0) + \frac{\eta}{mn} \cdot \rho_r(t)v_r(0)\left(1 - \frac{1}{m\rho_r(t)}\mathbf{w}_r^\top(t)\mathbf{X}^\top \mathbf{X}\mathbf{W}(t)\mathbf{v}(t)\right) + \frac{\eta}{mn}\langle \mathbf{w}_r(0), \mathbf{X}^\top \mathbf{y}\rangle v_r(0)$$

$$= v_r(t)v_r(0) + \frac{\eta \rho_r(t)v_r(0)}{mn}\left(1 - \frac{1}{m\rho_r(t)}\sum_{r'=1}^m \mathbf{w}_r^\top(t)\mathbf{X}^\top \mathbf{X}\mathbf{w}_{r'}(t)v_{r'}(t)\right) + \frac{\eta}{mn}\langle \mathbf{w}_r(0), \mathbf{X}^\top \mathbf{y}\rangle v_r(0)$$

$$\leq v_r(t)v_r(0) + \frac{\eta \rho_r(t)v_r(0)}{mn}\left(1 + \frac{1}{m\rho_r(t)}\sum_{r'=1}^m 1/4(\mathbf{w}_r^\top(t) - \mathbf{w}_{r'}^\top(t))\mathbf{X}^\top \mathbf{X}(\mathbf{w}_r^\top(t) - \mathbf{w}_{r'}^\top(t))\right)v_{r'}(t))$$

$$+ \frac{\eta}{mn}\langle \mathbf{w}_r(0), \mathbf{X}^\top \mathbf{y}\rangle v_r(0)$$

$$\leq v_r(t)v_r(0) + \frac{\eta \rho_r(t)v_r(0)}{mn}\left(1 + \frac{2n}{m\rho_r(t)}\sum_{r'=1}^m 1/4\|\mathbf{w}_r^\top(t) - \mathbf{w}_{r'}^\top(t)\|_{\mathbf{\Sigma}}^2 v_{r'}(t)\right) + \frac{\eta\langle \mathbf{w}_r(0), \mathbf{X}^\top \mathbf{y}\rangle v_r(0)}{mn}$$

$$\leq v_r(t)v_r(0) + \frac{\eta \rho_r(t)v_r(0)}{mn}\left(1 + \frac{2n\|\mathbf{\Sigma}\|_2}{m\rho_r(t)\sqrt{\gamma}}\sum_{r'=1}^m 1/4\|\mathbf{w}_r^\top(t) - \mathbf{w}_{r'}^\top(t)\|_2^2\right) + \frac{\eta\langle \mathbf{w}_r(0), \mathbf{X}^\top \mathbf{y}\rangle v_r(0)}{mn}$$

$$\leq v_r(t)v_r(0) + \frac{\eta}{mn} \cdot \rho_r(t)v_r(0)\left(1 + 0.1\frac{\mathrm{tr}(\mathbf{\Sigma}) + \epsilon_\lambda}{\gamma^{3/2}}\right).$$

The first inequality is by the polarization identity $\boldsymbol{\alpha}^\top \mathbf{A}\boldsymbol{\beta} = 1/4(\boldsymbol{\alpha} + \boldsymbol{\beta})^\top \mathbf{A}(\boldsymbol{\alpha} + \boldsymbol{\beta}) - 1/4(\boldsymbol{\alpha} - \boldsymbol{\beta})^\top \mathbf{A}(\boldsymbol{\alpha} - \boldsymbol{\beta})$. The second inequality is by Lemma A.3. The third inequality is by quadratic form expansion. The final inequality is by the condition that $\mathrm{tr}(\mathbf{\Sigma}) \geq n\|\mathbf{\Sigma}\|_2$.

**Fifth claim**. Finally, we complete the proof of the fifth induction claim for $\|\mathbf{w}_r(t) - \mathbf{w}_r(0) - \rho_r(t)/\gamma \cdot \mathbf{X}^\top \mathbf{y}\|_2$:

$$\left\|\mathbf{w}_r(t+1) - \mathbf{w}_r(0) - \rho_r(t+1)/\gamma \cdot \mathbf{X}^\top \mathbf{y}\right\|_2$$

$$= \left\|\mathbf{w}_r(t) - \mathbf{w}_r(0) - \rho_r(t+1)/\gamma \mathbf{X}^\top \mathbf{y} + \frac{\eta}{mn} \cdot v_r(t)\mathbf{X}^\top \mathbf{y} - \frac{\eta}{m^2 n} \cdot v_r(t)\mathbf{X}^\top \mathbf{X}\mathbf{W}(t)\mathbf{v}(t)\right\|_2$$

$$= \left\|\mathbf{w}_r(t) - \mathbf{w}_r(0) - \left(\rho_r(t) + \frac{\eta}{nm}v_r(t)\gamma - \frac{\eta}{m^2 n} \cdot v_r(t) \cdot \mathbf{y}^\top \mathbf{X}\mathbf{X}^\top \mathbf{X}\mathbf{W}(t)\mathbf{v}(t)\right)/\gamma \cdot \mathbf{X}^\top \mathbf{y}\right.$$

$$\left.+ \frac{\eta}{mn} \cdot v_r(t) \cdot \left(\mathbf{X}^\top \mathbf{y} - \frac{1}{m}\mathbf{X}^\top \mathbf{X}\mathbf{W}(t)\mathbf{v}(t)\right)\right\|_2$$

$$\leq \left\|\mathbf{w}_r(t) - \mathbf{w}_r(0) - \rho_r(t)/\gamma \mathbf{X}^\top \mathbf{y}\right\|_2 + \frac{\eta v_r(t)}{m^2 n}\left\|\mathbf{y}^\top \mathbf{X}\mathbf{X}^\top \mathbf{X}\mathbf{W}(t)\mathbf{v}(t)/\gamma \cdot \mathbf{X}^\top \mathbf{y} - \mathbf{X}^\top \mathbf{X}\mathbf{W}(t)\mathbf{v}(t)\right\|_2$$

$$\leq \left\|\mathbf{w}_r(t) - \mathbf{w}_r(0) - \rho_r(t)/\gamma \mathbf{X}^\top \mathbf{y}\right\|_2 + \frac{\eta v_r(t)}{m^2 n}\left\|\mathbf{X}^\top \mathbf{y}\mathbf{y}^\top \mathbf{X}\mathbf{X}^\top \mathbf{X}/\gamma - \mathbf{X}^\top \mathbf{X}\right\|_2 \|\mathbf{W}(t)\mathbf{v}(t)\|_2$$

$$\leq \left\|\mathbf{w}_r(t) - \mathbf{w}_r(0) - \rho_r(t)/\gamma \cdot \mathbf{X}^\top \mathbf{y}\right\|_2 + \frac{\eta}{m^2 n} \cdot v_r(t) \cdot \|\mathbf{X}^\top \mathbf{y}\mathbf{y}^\top \mathbf{X}/\gamma - \mathbf{I}\|_2 \|\mathbf{X}^\top \mathbf{X}\|_2 \|\mathbf{W}(t)\mathbf{v}(t)\|_2$$

$$\leq 1.5\frac{(\mathrm{tr}(\mathbf{\Sigma}) + \epsilon_\lambda)\rho_r(t)}{\gamma^2} + \frac{2(\epsilon_\lambda + \mathrm{tr}(\mathbf{\Sigma}))m\eta v_r^2(t)}{m^2 n}\left(1.5\frac{\mathrm{tr}(\mathbf{\Sigma}) + \epsilon_\lambda}{\gamma^{3/2}}\rho_r(t)/\sqrt{\gamma} + \rho_r(t)/\sqrt{\gamma} + \|\mathbf{w}_r(0)\|_2\right)$$

$$\leq 1.5\frac{\mathrm{tr}(\mathbf{\Sigma}) + \epsilon_\lambda}{\gamma^{3/2}}/\sqrt{\gamma}\left(\rho_r(t) + \frac{\eta}{mn} \cdot v_r(t)(1 - 1.1\frac{\mathrm{tr}(\mathbf{\Sigma}) + \epsilon_\lambda}{\gamma^{3/2}})\right)$$

$$\leq 1.5\frac{\mathrm{tr}(\mathbf{\Sigma}) + \epsilon_\lambda}{\gamma^{3/2}}\rho_r(t+1)/\sqrt{\gamma}.$$

The first inequality is by triangle inequality. The second and third inequalities are by Cauchy–Schwarz inequality. The forth inequality is by triangle inequality.

At the end of stage 1, we check the distance between neural tangent kernel of the two-layer neural network and target-aligned kernel:

$$\left\| \mathbf{\Theta}(\mathbf{X}, \mathbf{X}, t_1) - \frac{1}{m\gamma}\mathbf{X}\left(\mathbf{I} + \frac{1}{\gamma}\mathbf{X}^\top\mathbf{yy}^\top\mathbf{X}\right)\mathbf{X}^\top \right\|_2$$

$$\leq \left\| \frac{1}{m^2}\sum_{r=1}^m v_r^2(t_1)\mathbf{XX}^\top - \frac{1}{m\gamma}\mathbf{XX}^\top \right\|_2 + \left\| \frac{1}{m^2}\sum_{r=1}^m \mathbf{Xw}_r(t_1)\mathbf{w}_r^\top(t_1)\mathbf{X}^\top - \frac{1}{m\gamma^2}\mathbf{XX}^\top\mathbf{yy}^\top\mathbf{XX}^\top \right\|_2$$

$$\leq C\frac{1}{m}\left(\frac{\epsilon_\lambda + \operatorname{tr}(\mathbf{\Sigma})}{\gamma^{3/2}}\right)^2 + \frac{1}{m}\left\| \frac{1}{m}\sum_{r=1}^m \mathbf{Xw}_r(t_1)\mathbf{w}_r^\top(t_1)\mathbf{X}^\top - \frac{1}{m\gamma}\sum_{r=1}^m \mathbf{Xw}_r(t_1)\mathbf{y}^\top\mathbf{XX}^\top \right\|_2$$

$$+ \frac{1}{m}\left\| \frac{1}{m\gamma}\sum_{r=1}^m \mathbf{Xw}_r(t_1)\mathbf{y}^\top\mathbf{XX}^\top - \frac{1}{m\gamma^2}\mathbf{XX}^\top\mathbf{yy}^\top\mathbf{XX}^\top \right\|_2$$

$$= C\frac{1}{m}\left(\frac{\epsilon_\lambda + \operatorname{tr}(\mathbf{\Sigma})}{\gamma^{3/2}}\right)^2 + \frac{1}{m}\left\| \frac{1}{m}\sum_{r=1}^m \mathbf{Xw}_r(t_1)(\mathbf{w}_r^\top(t) - \mathbf{y}^\top\mathbf{X}/\gamma)\mathbf{X}^\top \right\|_2$$

$$+ \frac{1}{m}\left\| \frac{1}{m\gamma}\mathbf{X}\sum_{r=1}^m \left(\mathbf{w}_r(t_1) - \mathbf{X}^\top\mathbf{y}\right)\mathbf{y}^\top\mathbf{XX}^\top \right\|_2$$

$$= \frac{C}{m}\left(\frac{\epsilon_\lambda + \operatorname{tr}(\mathbf{\Sigma})}{\gamma^{3/2}}\right)^2 + \frac{\sum_{r=1}^m \mathbf{w}_r^\top(t_1)\mathbf{X}^\top\mathbf{X}\left(\mathbf{w}_r(t) - \frac{\mathbf{X}^\top\mathbf{y}}{\gamma}\right)}{m^2} + \frac{\sum_{r=1}^m \mathbf{y}^\top\mathbf{XX}^\top\mathbf{X}\left(\mathbf{w}_r(t_1) - \mathbf{X}^\top\mathbf{y}/\gamma\right)}{m^2\gamma}$$

$$\leq C\frac{1}{m}\left(\frac{\epsilon_\lambda + \operatorname{tr}(\mathbf{\Sigma})}{\gamma^{3/2}}\right)^2 + \frac{\epsilon_\lambda + \operatorname{tr}(\mathbf{\Sigma})}{m}(\|\mathbf{w}_r(0)\|_2 + \rho_r(t_1)/\sqrt{\gamma})(\|\mathbf{w}_r(0)\|_2 + 1.5\frac{\epsilon_\lambda + \operatorname{tr}(\mathbf{\Sigma})}{\gamma^{3/2}}\rho_r(t_1)/\sqrt{\gamma})$$

$$+ \frac{\epsilon_\lambda + \operatorname{tr}(\mathbf{\Sigma})}{m\gamma}(\sqrt{\gamma})(\|\mathbf{w}_r(0)\|_2 + 1.5\frac{\epsilon_\lambda + \operatorname{tr}(\mathbf{\Sigma})}{\gamma^{3/2}}\rho_r(t_1)/\sqrt{\gamma})$$

$$\leq C\frac{1}{m}\left(\frac{\epsilon_\lambda + \operatorname{tr}(\mathbf{\Sigma})}{\gamma^{3/2}}\right)^2.$$

The first inequality is by triangle inequality. The second inequality is by the first and the second claim in Equation (13). Finally, by Lemma A.4, we achieve the final result. $\qquad\square$

After the first stage, the last term in the gradient descent cannot be neglected. Consequently, we construct the following lemma to elucidate the full dynamics in the second stage:

**Lemma B.2** (Restatement of Lemma 5.3). *Under Assumption 4.1, there exist a time step $t_2$ satisfying $t_2 - t_1 = \Omega\left(\frac{m\sqrt{n}}{\eta\|\overline{\boldsymbol{\beta}}\|_2\sqrt{\operatorname{tr}(\mathbf{\Sigma})}}\log(\frac{n\|\overline{\boldsymbol{\beta}}\|_2\operatorname{tr}(\mathbf{\Sigma})}{\epsilon})\right)$ such that $L(f(\mathbf{X}, t_2), \mathbf{y}) \leq 4\epsilon$. Besides, during the second stage $t_1 \leq t \leq t_2$, we have:*

- $\|\mathbf{w}_r(t) - \mathbf{w}_r(0) - \rho(t)\mathbf{X}^\top\mathbf{y}\|_2 \leq \frac{1}{2}\rho_r(t)/\sqrt{\operatorname{tr}(\mathbf{\Sigma})n\|\overline{\boldsymbol{\beta}}\|_2^2}$.

- $|\rho_r(t) - v_r(t)\sqrt{\gamma}| \leq \frac{1}{2}\rho_r(t)$.

- $\left\| \mathbf{\Theta}(\mathbf{X}, \mathbf{X}, t) - v_r^2(t)\mathbf{X}\left(\mathbf{I} + \frac{1}{\gamma}\mathbf{X}^\top\mathbf{yy}^\top\mathbf{X}\right)\mathbf{X}^\top \right\|_2 \leq C\frac{v_r^2(t)\operatorname{tr}(\mathbf{\Sigma})}{m}$.

*Proof of Lemma B.2.* We adopt an induction method to show during the first stage $t_1 \leq t \leq t_2$, where $t_1 = \frac{1}{2}\frac{C\log(1/v_0)mn}{\eta\sqrt{\gamma}}$, and $t_2 = t_1 + \frac{nm}{\eta\sqrt{\gamma}}\log\left(\sqrt{\frac{1-\epsilon_\rho}{\epsilon_\rho\frac{\operatorname{tr}(\mathbf{\Sigma})}{\gamma^{3/2}}}}\right)$, with $\epsilon_\rho = 1 - \frac{\operatorname{tr}(\mathbf{\Sigma})}{\gamma^{3/2}}\rho_r^2(t_2)$ and probabilities at least $1 - n^{-2}$, the following inequalities hold for all $r \in [m]$:

- $\|\mathbf{w}_r(t) - \mathbf{w}_r(0) - \rho_r(t)/\gamma\mathbf{X}^\top\mathbf{y}\| \leq \frac{1}{2}\rho_r(t)/\sqrt{\gamma}$.

- $|\rho_r(t) - v_r(t)\sqrt{\gamma}| \leq \frac{1}{2}\rho_r(t)$.

- $\tilde{\rho}_r(t+1) \le \frac{3}{2}\tilde{\rho}_r(t)(1 + \frac{3}{2}\frac{\eta\sqrt{\gamma}}{mn})$.

- $\tilde{\rho}_r(t+1) \ge \tilde{\rho}_r(t)(1 + \frac{1}{4}\frac{\eta\sqrt{\gamma}}{mn})$.

It is not hard to check that all the inequalities hold at $t = t_1$. We then assume that all the inequalities hold at the time step of $t_1 < t < t_2$. By the induction method, we have the following derivation:

**First claim**:

$$\left\|\mathbf{w}_r(t+1) - \mathbf{w}_r(0) - \rho_r(t+1)/\gamma \cdot \mathbf{X}^\top\mathbf{y}\right\|_2$$

$$= \left\|\mathbf{w}_r(t) - \mathbf{w}_r(0) - \rho_r(t+1)/\gamma\mathbf{X}^\top\mathbf{y} + \frac{\eta}{mn}v_r(t)\mathbf{X}^\top\mathbf{y} - \frac{\eta}{m^2n}v_r(t)\mathbf{X}^\top\mathbf{X}\mathbf{W}(t)\mathbf{v}(t)\right\|_2$$

$$\le \left\|\mathbf{w}_r(t) - \mathbf{w}_r(0) - \rho_r(t)/\gamma \cdot \mathbf{X}^\top\mathbf{y}\right\|_2 + \frac{\eta v_r(t)}{m^2n}\left\|\mathbf{y}^\top\mathbf{X}\mathbf{X}^\top\mathbf{X}\mathbf{W}(t)\mathbf{v}(t)/\gamma \cdot \mathbf{X}^\top\mathbf{y} - \mathbf{X}^\top\mathbf{X}\mathbf{W}(t)\mathbf{v}(t)\right\|_2$$

$$\le \left\|\mathbf{w}_r(t) - \mathbf{w}_r(0) - \rho_r(t)/\gamma \cdot \mathbf{X}^\top\mathbf{y}\right\|_2 + \frac{\eta v_r(t)}{m^2n}\left\|\mathbf{X}^\top\mathbf{y}\mathbf{y}^\top\mathbf{X}\mathbf{X}^\top\mathbf{X}\mathbf{W}(t)\mathbf{v}(t)/\gamma - \mathbf{X}^\top\mathbf{X}\mathbf{W}(t)\mathbf{v}(t)\right\|_2$$

$$\le \left\|\mathbf{w}_r(t) - \mathbf{w}_r(0) - \rho_r(t)/\gamma \cdot \mathbf{X}^\top\mathbf{y}\right\|_2 + \frac{\eta}{m^2n}\cdot v_r(t)\cdot 2\left\|\mathbf{X}^\top\mathbf{X}\mathbf{W}(t)\mathbf{v}(t)\right\|_2$$

$$\le \left\|\mathbf{w}_r(t) - \mathbf{w}_r(0) - \rho_r(t)/\gamma \cdot \mathbf{X}^\top\mathbf{y}\right\|_2 + \frac{1}{2}\frac{\eta}{mn}\cdot v_r(t)(\sqrt{\gamma}/C - \langle\mathbf{X}^\top\mathbf{X}\mathbf{W}(t)\mathbf{v}(t), \mathbf{X}^\top\mathbf{y}\rangle)$$

$$\le \frac{1}{2}\rho_r(t+1)/\sqrt{\gamma}.$$

In the last second inequality we have used the following inequality:

$$\frac{1}{m}\left\|\mathbf{X}^\top\mathbf{X}\mathbf{W}(t)\mathbf{v}(t)\right\|_2 \le C\sqrt{\gamma} - \frac{1}{m}\langle\mathbf{X}^\top\mathbf{X}\mathbf{W}(t)\mathbf{v}(t), \mathbf{X}^\top\mathbf{y}/\sqrt{\gamma}\rangle.$$

We then confirm the above inequality:

$$\frac{1}{m}\langle\mathbf{X}^\top\mathbf{X}\mathbf{W}(t)\mathbf{v}(t), \mathbf{X}^\top\mathbf{y}/\sqrt{\gamma}\rangle + \frac{1}{m}\left\|\mathbf{X}^\top\mathbf{X}\mathbf{W}(t)\mathbf{v}(t)\right\|_2 \le \frac{2}{m}\left\|\mathbf{X}^\top\mathbf{X}\mathbf{W}(t)\mathbf{v}(t)\right\|_2$$

$$\le \frac{2\|\mathbf{X}^\top\mathbf{X}\|_2}{m}\left\|\sum_{r=1}^m \mathbf{w}(t)v_r(t)\right\|_2 \le \frac{2\|\mathbf{X}^\top\mathbf{X}\|_2}{m}\sum_{r=1}^m\|\mathbf{w}(t)\|_2 v_r(t) \le \frac{2\|\mathbf{X}^\top\mathbf{X}\|_2}{m}\sum_{r=1}^m\|\mathbf{w}_r(t_2)\|_2 v_r(t_2)$$

$$\le 3\|\mathbf{X}^\top\mathbf{X}\|_2\rho_r(t_2)v_r(t_2)/\sqrt{\gamma} \le \frac{9}{2}\|\mathbf{X}^\top\mathbf{X}\|_2\rho_r^2(t_2)/\gamma \le \frac{9}{2}\sqrt{\gamma}.$$

**Second claim:**

$$\left|\rho_r(t+1) - v_r(t+1)\sqrt{\gamma}\right|$$

$$= \left|\rho_r(t) + \frac{\eta}{mn}\cdot v_r(t)\cdot\gamma - \frac{\eta}{m^2n}\cdot v_r(t)\cdot\langle\mathbf{X}^\top\mathbf{X}\mathbf{W}(t)\mathbf{v}(t), \mathbf{X}^\top\mathbf{y}\rangle\right.$$

$$\left. - v_r(t)\sqrt{\gamma} - \frac{\eta\sqrt{\gamma}}{mn}\cdot\rho_r(t) - \frac{\eta\sqrt{\gamma}}{mn}\cdot\langle\mathbf{w}_r(0), \mathbf{X}^\top\mathbf{y}\rangle + \frac{\eta\sqrt{\gamma}}{m^2n}\cdot\langle\mathbf{w}_r(t), \mathbf{X}^\top\mathbf{X}\mathbf{W}(t)\mathbf{v}(t)\rangle\right|$$

$$= \left|(\rho_r(t) - v_r(t)\sqrt{\gamma}) - \frac{\eta\sqrt{\gamma}}{mn}(\rho_r(t) - v_r(t)\sqrt{\gamma}) + \frac{\eta\sqrt{\gamma}}{m^2n}\left(\langle\mathbf{w}_r(t) - v_r(t)\mathbf{X}^\top\mathbf{y}/\sqrt{\gamma}, \mathbf{X}^\top\mathbf{X}\mathbf{W}(t)\mathbf{v}(t)\rangle\right)\right|$$

$$\le \left|(\rho_r(t) - v_r(t)\sqrt{\gamma}) - \frac{\eta\sqrt{\gamma}}{mn}(\rho_r(t) - v_r(t)\sqrt{\gamma}) + \frac{\eta\sqrt{\gamma}}{m^2n}\left\|\mathbf{w}_r(t) - v_r(t)\mathbf{X}^\top\mathbf{y}/\sqrt{\gamma}\right\|_2\left\|\mathbf{X}^\top\mathbf{X}\mathbf{W}(t)\mathbf{v}(t)\right\|_2\right|$$

$$\le \left|\left(1 - \frac{\eta\sqrt{\gamma}}{mn}\right)(\rho_r(t) - v_r(t)\sqrt{\gamma}) + \frac{\eta\sqrt{\gamma}}{m^2n}\sqrt{\gamma}(|\rho_r(t)/\sqrt{\gamma} - v_r(t)|\right.$$

$$\left. + \|\mathbf{w}_r(t) - \mathbf{w}_r(0) - \rho_r(t)/\gamma\mathbf{X}^\top\mathbf{y}\|_2 + \|\mathbf{w}_r(0)\|_2)\left\|\mathbf{X}^\top\mathbf{X}\mathbf{W}(t)\mathbf{v}(t)\right\|_2\right|$$

$$\le |(\rho_r(t) - v_r(t)\sqrt{\gamma})| + 1.1\frac{\eta\sqrt{\gamma}}{m^2n}\|\mathbf{w}_r(t) - \mathbf{w}_r(0) - \rho_r(t)/\gamma\mathbf{X}^\top\mathbf{y}\|_2\left\|\mathbf{X}^\top\mathbf{X}\mathbf{W}(t)\mathbf{v}(t)\right\|_2$$

$$\le |(\rho_r(t) - v_r(t)\sqrt{\gamma})| + \frac{1.1}{2}\frac{\eta\sqrt{\gamma}}{m^2n}\rho_r(t)\left\|\mathbf{X}^\top\mathbf{X}\mathbf{W}(t)\mathbf{v}(t)\right\|_2$$

$$\le \frac{1}{2}\rho_r(t+1).$$

The first inequality is by Cauchy–Schwarz inequality. The second and third inequalities are by the third induction claim and triangle inequality. The fourth inequality is by the second induction claim.

**Third claim**

we define $\tilde{\rho}_r(t) = \sqrt{\frac{\rho_r^2(t)}{1-\frac{\text{tr}(\mathbf{\Sigma})}{\gamma^{3/2}}\rho_r^2(t)}}$, then the update rule of $\tilde{\rho}_r(t)$ can be expressed as follows:

$$
\begin{aligned}
\tilde{\rho}_r(t+1) &= \sqrt{\frac{\rho_r^2(t+1)}{1-\frac{\text{tr}(\mathbf{\Sigma})}{\gamma^{3/2}}\rho_r^2(t+1)}} \\
&\leq \frac{\rho_r(t)+\frac{3}{2}\frac{\eta\sqrt{\gamma}}{nm}\rho_r(t)\left(1-\frac{1}{16}\frac{\text{tr}(\mathbf{\Sigma})}{\gamma^{3/2}}\rho_r^2(t)\right)}{\sqrt{1-\frac{\text{tr}(\mathbf{\Sigma})}{\gamma^{3/2}}\left(\rho_r(t)+\frac{\eta\sqrt{\gamma}}{nm}\rho_r(t)(\frac{3}{2}-\frac{1}{16}\frac{\text{tr}(\mathbf{\Sigma})}{\gamma^{3/2}}\rho_r^2(t))\right)^2}} \\
&\leq \frac{3}{2}\tilde{\rho}_r(t)+\frac{9}{4}\frac{\eta\sqrt{\gamma}}{nm}\left(1-\frac{1}{2}\frac{\text{tr}(\mathbf{\Sigma})}{\gamma^{3/2}}\rho_r^2(t)\right)\tilde{\rho}_r(t) \\
&\leq \frac{3}{2}\tilde{\rho}_r(t)+\frac{9}{4}\frac{\eta\sqrt{\gamma}}{nm}\tilde{\rho}_r(t).
\end{aligned}
$$

where we have used the upper bound for $\rho_r(t)$:

$$
\begin{aligned}
\rho_r(t+1) &= \rho_r(t)+\frac{\eta}{mn}\cdot v_r(t)\cdot\gamma-\frac{\eta}{m^2 n}\cdot v_r(t)\cdot\langle\mathbf{X}^\top\mathbf{X}\mathbf{W}(t)\mathbf{v}(t),\mathbf{X}^\top\mathbf{y}\rangle \\
&\leq \rho_r(t)+\frac{3}{2}\frac{\eta}{mn}\cdot\rho_r(t)\cdot\sqrt{\gamma}-\frac{\eta}{mn}\cdot\frac{1}{8}\rho_r^3(t)/\gamma\cdot(\text{tr}(\mathbf{\Sigma})-\epsilon_\lambda) \\
&\leq \rho_r(t)+\frac{3}{2}\frac{\eta}{mn}\cdot\rho_r(t)\cdot\sqrt{\gamma}-\frac{1}{16}\frac{\eta}{mn}\cdot\rho_r^3(t)/\gamma\cdot\text{tr}(\mathbf{\Sigma}),
\end{aligned}
$$

where last inequality is by our assumption that $\epsilon_\lambda < \text{tr}(\mathbf{\Sigma})$. Besides, we show the lower bound for $\rho_r(t)$ as follows:

$$
\begin{aligned}
\rho_r(t+1) &= \rho_r(t)+\frac{\eta}{mn}\cdot v_r(t)\cdot\gamma-\frac{\eta}{m^2 n}\cdot v_r(t)\cdot\langle\mathbf{X}^\top\mathbf{X}\mathbf{W}(t)\mathbf{v}(t),\mathbf{X}^\top\mathbf{y}\rangle \\
&\geq \rho_r(t)+\frac{1}{2}\frac{\eta}{mn}\cdot\rho_r(t)\cdot\sqrt{\gamma}-\frac{\eta}{mn}\cdot\frac{27}{8}\rho_r^3(t)/\gamma\cdot(\text{tr}(\mathbf{\Sigma})+\epsilon_\lambda) \\
&\geq \rho_r(t)+\frac{1}{2}\frac{\eta}{mn}\cdot\rho_r(t)\cdot\sqrt{\gamma}-\frac{81}{16}\frac{\eta}{m^2 n}\cdot\rho_r^3(t)/\gamma\cdot\text{tr}(\mathbf{\Sigma}).
\end{aligned}
$$

**Fourth claim**

$$
\begin{aligned}
\tilde{\rho}_r(t+1) &= \sqrt{\frac{\rho_r^2(t+1)}{1-\frac{\text{tr}(\mathbf{\Sigma})}{\gamma^{3/2}}\rho_r^2(t+1)}} \\
&\geq \frac{\rho_r(t)+\frac{1}{2}\frac{\eta\sqrt{\gamma}}{nm}\rho_r(t)\left(1-\frac{81}{16}\frac{\text{tr}(\mathbf{\Sigma})}{\gamma^{3/2}}\rho_r^2(t)\right)}{\sqrt{1-\frac{\text{tr}(\mathbf{\Sigma})}{\gamma^{3/2}}\left(\rho_r(t)+\frac{\eta\sqrt{\gamma}}{nm}\rho_r(t)(\frac{1}{2}-\frac{81}{16}\frac{\text{tr}(\mathbf{\Sigma})}{\gamma^{3/2}}\rho_r^2(t))\right)^2}} \\
&\geq \tilde{\rho}_r(t)+\frac{1}{2}\frac{\eta\sqrt{\gamma}}{nm}\left(1-\frac{81}{16}\frac{\text{tr}(\mathbf{\Sigma})}{\gamma^{3/2}}\rho_r^2(t)\right)\tilde{\rho}_r(t) \\
&\geq \tilde{\rho}_r(t)+\frac{1}{4}\frac{\eta\sqrt{\gamma}}{nm}\tilde{\rho}_r(t).
\end{aligned}
$$

For $t_1 \le t \le t_2$, we have

$$\left\| \Theta(\mathbf{X}, \mathbf{X}, t) - \frac{v_r^2(t)}{m} \mathbf{X} \left( \mathbf{I} + \frac{1}{\gamma} \mathbf{X}^\top \mathbf{y} \mathbf{y}^\top \mathbf{X} \right) \mathbf{X}^\top \right\|_2$$

$$\le \left\| \frac{1}{m^2} \sum_{r=1}^m v_r^2(t) \mathbf{X}\mathbf{X}^\top - \frac{v_{r'}^2(t)}{m} \mathbf{X}\mathbf{X}^\top \right\|_2 + \left\| \frac{1}{m^2} \sum_{r=1}^m \mathbf{X}\mathbf{w}_r(t)\mathbf{w}_r^\top(t)\mathbf{X}^\top - \frac{v_{r'}^2(t)}{m\gamma} \mathbf{X}\mathbf{X}^\top \mathbf{y}\mathbf{y}^\top \mathbf{X}\mathbf{X}^\top \right\|_2$$

$$\le \left| \frac{1}{m} \sum_{r=1}^m v_r^2(t) - v_{r'}^2(t) \right| \frac{\epsilon_\lambda + \mathrm{tr}(\mathbf{\Sigma})}{m} + \left\| \frac{1}{m^2} \sum_{r=1}^m \mathbf{X}\mathbf{w}_r(t)\mathbf{w}_r^\top(t)\mathbf{X}^\top - \frac{v_{r'}(t)}{m^2\sqrt{\gamma}} \sum_{r=1}^m \mathbf{X}\mathbf{w}_r(t)\mathbf{y}^\top \mathbf{X}\mathbf{X}^\top \right\|_2$$

$$+ \left\| \frac{v_{r'}(t)}{m^2\sqrt{\gamma}} \sum_{r=1}^m \mathbf{X}\mathbf{w}_r(t)\mathbf{y}^\top \mathbf{X}\mathbf{X}^\top - \frac{v_{r'}^2(t)}{m^2\gamma} \mathbf{X}\mathbf{X}^\top \mathbf{y}\mathbf{y}^\top \mathbf{X}\mathbf{X}^\top \right\|_2$$

$$= \left| \frac{1}{m} \sum_{r=1}^m v_r^2(t) - v_{r'}^2(t) \right| \frac{\epsilon_\lambda + \mathrm{tr}(\mathbf{\Sigma})}{m} + \left\| \frac{1}{m^2} \sum_{r=1}^m \mathbf{X}\mathbf{w}_r(t)(\mathbf{w}_r^\top(t) - v_{r'}(t)\mathbf{y}^\top \mathbf{X}/\sqrt{\gamma})\mathbf{X}^\top \right\|_2$$

$$+ \left\| \frac{v_{r'}(t)}{m^2\sqrt{\gamma}} \mathbf{X} \sum_{r=1}^m \left( \mathbf{w}_r(t) - v_{r'}(t)\mathbf{X}^\top \mathbf{y}/\sqrt{\gamma} \right) \mathbf{y}^\top \mathbf{X}\mathbf{X}^\top \right\|_2$$

$$\le C_1 \frac{v_{r'}^2(t)\mathrm{tr}(\mathbf{\Sigma})}{m} + \frac{1}{m^2} \sum_{r=1}^m \mathbf{w}_r^\top(t)\mathbf{X}^\top \mathbf{X} \left( \mathbf{w}_r(t) - v_{r'}(t)\mathbf{X}^\top \mathbf{y}/\sqrt{\gamma} \right)$$

$$+ \frac{v_{r'}^2(t)}{m^2\sqrt{\gamma}} \sum_{r=1}^m \mathbf{y}^\top \mathbf{X}\mathbf{X}^\top \mathbf{X} \left( \mathbf{w}_r(t) - v_{r'}(t)\mathbf{X}^\top \mathbf{y}/\sqrt{\gamma} \right)$$

$$\le C_1 \frac{v_r^2(t)\mathrm{tr}(\mathbf{\Sigma})}{m} + C_2 \frac{\mathrm{tr}(\mathbf{\Sigma})v_r^2(t)}{m} + C_3 \frac{\mathrm{tr}(\mathbf{\Sigma})v_r^2(t)}{m}$$

$$= C \frac{v_r^2(t)\mathrm{tr}(\mathbf{\Sigma})}{m}.$$

Denote that $\epsilon_\rho = 1 - \frac{\mathrm{tr}(\mathbf{\Sigma})}{\gamma^{3/2}} \rho_r^2(t_2)$. Finally, we calculate the training loss at time step $t_2$:

$$L(t_2) = \frac{1}{n} \left\| \frac{1}{m} \sum_{r=1}^m v_r(t_2)\mathbf{X}\mathbf{w}_r(t_2) - \mathbf{y} \right\|_2^2$$

$$\le \frac{1}{n} \left\| \frac{\rho_r^2(t_2)}{\gamma^{3/2}} \mathbf{X}\mathbf{X}^\top \mathbf{y} - \mathbf{y} \right\|_2^2 + \frac{1}{n} \left\| \mathbf{X}\mathbf{w}_r(0) \right\|_2^2 v_r^2(t_2)$$

$$\le \frac{1}{n} \left\| \frac{\rho_r^2(t_2)}{\gamma^{3/2}} \mathbf{X}\mathbf{X}^\top - \mathbf{I} \right\|_2^2 \|\mathbf{y}\|_2^2 + \frac{1}{n} \left\| \mathbf{X}\mathbf{w}_r(0) \right\|_2^2 v_r^2(t_2)$$

$$\le \frac{1}{n} \frac{\epsilon_\lambda^2}{\mathrm{tr}(\mathbf{\Sigma})^2} \|\mathbf{y}\|_2^2 + \frac{1}{n} \left\| \epsilon_\rho \mathbf{X}\mathbf{X}^\top \right\|_2^2 \|\mathbf{y}\|_2^2 + \frac{1}{n} \left\| \mathbf{X}\mathbf{w}_r(0) \right\|_2^2 v_r^2(t_2)$$

$$\le 2\|\widetilde{\boldsymbol{\beta}}\|_2^2 \frac{\epsilon_\lambda^2}{\mathrm{tr}(\mathbf{\Sigma})^2} + \epsilon_\rho^2 \|\widetilde{\boldsymbol{\beta}}\|_2^2 \mathrm{tr}(\mathbf{\Sigma})^2 + \widetilde{O}(\sigma_0^2 \gamma^{1/2})$$

$$\le 4\epsilon.$$

Therefore we can claim that there exist a time $t_2 - t_1 = \frac{mn}{\eta\sqrt{\gamma}} \log\left( \sqrt{\frac{\gamma^{3/2}}{\epsilon_\rho \mathrm{tr}(\mathbf{\Sigma})}} \right)$ such that $\epsilon_\rho^2 \|\widetilde{\boldsymbol{\beta}}\|_2^2 \mathrm{tr}(\mathbf{\Sigma})^2 \le \epsilon$. As a result, we obtain that $t_2 - t_1 = \Omega\left( \frac{m\sqrt{n}}{\eta\|\overline{\boldsymbol{\beta}}\|_2 \sqrt{\mathrm{tr}(\mathbf{\Sigma})}} \log(\frac{n\|\overline{\boldsymbol{\beta}}\|_2 \mathrm{tr}(\mathbf{\Sigma})}{\epsilon}) \right)$. $\qquad\square$

