# OpenReview forum: "Exact and Rich Feature Learning Dynamics of Two-Layer Linear Networks"
_CPAL.cc/2025/Proceedings_Track — CPAL 2025 (Proceedings Track) Poster_

### Official Review · Reviewer_gtng · 2025-01-09
**Review of Submission86**

**Rating:** 5
**Confidence:** 3

**Review:**

Summary
This paper provides an analysis of feature learning dynamics for two layer linear networks by focusing on the dynamics of two quantities: feature alignment and network magnitude, which correspond to two distinct phases of feature learning.

Pros
The development of the analysis is clear and mathematically sound.
Experiments validate the theoretical claims quite cleanly.

Cons
I am unsure of the novelty of this work. Linear network dynamics have been studied quite extensively, and the “align then fit” dynamics have been studied and noted in prior works, even in nonlinear settings [1] [2]. The presented table with comparison to existing works is also not completely clear, what does “Full 2-stage feature learning” mean here? If the distinctions of this work from prior work could be made more clear, I would be inclined to increase my score.

[1] Maennel et al. Gradient descent quantizes relu network features.
[2] Boursier et al. Gradient flow dynamics of shallow relu networks for square loss and orthogonal inputs.

---

### Official Review · Reviewer_ZFwo · 2025-01-10
**Two Stages Feature Learning Dynamics in Two-layer Linear Network**

**Rating:** 7
**Confidence:** 3

**Review:**

The paper investigates the training dynamics of two-layer linear networks under gradient descent. With small initialization on the model weights, the authors demonstrate the training of the network undergoes a two-stage phenomenon where in the first stage, the network output is small to learn data features and the loss convergence mainly happens in the second stage.  The theoretical findings are supported by both synthetic and CIFAR-10 experiments.

**Pros:**
* The paper is clearly written and well-motivated, providing sufficient details to support the theoretical framework.
* The exploration of the two-stage training dynamic is both novel and insightful.
* The experiments effectively corroborate the theoretical results

**Cons:**
* I'm curious about the applicability of the results, does the two-stage phenomenon also happens for non-linear networks? Also, how does the observed behavior change if the initialization assumption is violated?
* Line 186, why is the right hand side negligible? Is this because of the assumed small initialization?
* What would be the practical indications of the theoretical results for modern deep neural network training?

---

### Official Review · Reviewer_tmCN · 2025-01-14
**Review for Submission86**

**Rating:** 8
**Confidence:** 3

**Review:**

**Summary**:
This paper investigates the training dynamics of two-layer linear neural networks under gradient descent. It provides a precise characterization of feature learning, proposing a two-stage dynamic system: an initial feature alignment phase followed by a magnitude amplification phase. By analyzing the non-linear dynamics of feature alignment and network magnitude, the study establishes a connection between feature learning and the evolution of the neural tangent kernel (NTK). The theoretical findings are validated through experiments on synthetic and real-world datasets, offering new insights into the non-asymptotic behavior of neural networks beyond the lazy training regime.

**Strengths**:
1. **Rigorous Theoretical Framework**: The paper provides a detailed non-asymptotic analysis of feature learning dynamics, a notable improvement over prior studies focused primarily on asymptotic or simplified settings. Moreover, The introduction of a two-stage dynamic system (feature alignment and magnitude amplification) offers a comprehensive understanding of the training trajectory.

2. **Novel Contributions**: The study connects feature learning dynamics to NTK evolution, demonstrating that networks undergo feature adaptation rather than static kernel behavior.The analysis captures the transition between alignment and interpolation phases, which is crucial for understanding the generalization ability of over-parameterized models.

3. **Empirical Validation**: Simulations on synthetic and real-world datasets (e.g., CIFAR-10) corroborate the theoretical findings, showcasing the practical relevance of the two-stage training dynamics.

4. **Comparison with Existing Work and clear presentation**:The paper situates its contributions within the broader literature on NTK and feature learning, clearly highlighting its advancements over existing methods. The inclusion of visualizations (e.g., training loss curves, NTK evolution) and comparisons with related works enhances the clarity and accessibility of the paper.

**Weaknesses**:
1. **Limited Applicability and Scalability Concerns**: The study focuses exclusively on two-layer linear networks, which, while analytically tractable, may not generalize to deeper or non-linear architectures commonly used in practice. The computational requirements of analyzing NTK evolution and feature alignment may become prohibitive for larger datasets or wider networks, limiting the scalability of the proposed approach.
2. **Simplified Assumptions**: Certain assumptions, such as sub-Gaussian noise or specific initialization conditions, might not hold in real-world scenarios, potentially restricting the broader applicability of the results.

---

### Meta-Review · Area_Chair_d1HU · 2025-02-06

**Recommendation:** Accept (Poster)
**Confidence:** 5

**Metareview:**

This paper presents a detailed theoretical analysis of the training dynamics in two-layer linear networks, introducing a novel two-stage framework for feature learning. The approach is well-supported by experiments on synthetic and real-world datasets, validating the proposed dynamics. All reviewers found the paper clear, mathematically sound, and a valuable contribution to understanding neural network training.

Based my evaluation and the reviewers' comments, I recommend acceptance for this paper.

---

### Decision · Program_Chairs · 2025-02-11

Accept (Poster)